# Deciphering the mechanism of glutaredoxin-catalyzed roGFP2 redox sensing reveals a ternary complex with glutathione for protein disulfide reduction

Fabian Geissel[1,3], Lukas Lang[1,3], Britta Husemann[1], Bruce Morgan [2] & Marcel Deponte [1] ✉

Glutaredoxins catalyze the reduction of disulfides and are key players in redox metabolism and regulation. While important insights were gained regarding the reduction of glutathione disulfide substrates, the mechanism of non-glutathione disulfide reduction remains highly debated. Here we determined the rate constants for the individual redox reactions between PfGrx, a model glutaredoxin from *Plasmodium falciparum*, and redox-sensitive green fluorescent protein 2 (roGFP2), a model substrate and versatile tool for intracellular redox measurements. We show that the PfGrx-catalyzed oxidation of roGFP2 occurs via a monothiol mechanism and is up to three orders of magnitude faster when roGFP2 and PfGrx are fused. The oxidation kinetics of roGFP2-PfGrx fusion constructs reflect at physiological GSSG concentrations the glutathionylation kinetics of the glutaredoxin moiety, thus allowing intracellular structure-function analysis. Reduction of the roGFP2 disulfide occurs via a monothiol mechanism and involves a ternary complex with GSH and PfGrx. Our study provides the mechanistic basis for understanding roGFP2 redox sensing and challenges previous mechanisms for protein disulfide reduction.

Class I glutaredoxins (Grx) are glutathione-dependent thiol:disulfide oxidoreductases that reversibly convert a variety of low or high molecular weight disulfide substrates using the cysteinyl-containing tripeptide glutathione (GSH) as a reducing agent (EC 1.8.4.1–4)[1–8]. The disulfide substrates of glutaredoxins can be subdivided into two major groups. The first group comprises glutathione disulfide substrates (GSSR) that are reduced by one molecule of GSH, yielding a thiol (RSH) and glutathione disulfide (GSSG) as products (Fig. 1a)[5–8]. Model and physiological GSSR substrates are, for example, L-cysteine-glutathione disulfide (GSSCys)[9–15], coenzyme A-glutathione disulfide[3,16], as well as S-glutathionylated serum albumin[9,11,17–20], sirtuin[21], sulfiredoxin[22], and peroxiredoxins[23–29]. The second group comprises non-glutathione disulfide substrates (RS₂ or RSSR') that are reduced by two molecules of GSH, yielding one dithiol product or two monothiol products (R(SH)₂ or RSH + R'SH) and GSSG (Fig. 1a)[6–8]. Model and physiological RS₂ and RSSR' substrates are, for example, bis(2-hydroxyethyl)disulfide (HEDS)[1,3,12,14,30], L-cystine[1,3], redox-sensitive yellow or green fluorescent protein 2 (rxYFP or roGFP2)[14,15,31–35], insulin[36,37], collapsin response mediator protein 2[38,39], the transcription factor OxyR[40], 3'-phosphoadenosine 5'-phosphosulfate reductase (PR)[41,42], methionine sulfoxide reductase (MSR)[34,43,44], and ribonucleotide reductase (RNR)[2,34,45–47]. Glutaredoxins are therefore key enzymes for redox catalysis and physiological processes such as DNA synthesis, signal transduction, and organ development[2,21,34,38,40,47,48]. While important

---

[1]Faculty of Chemistry, Comparative Biochemistry, RPTU Kaiserslautern, D-67663 Kaiserslautern, Germany. [2]Institute of Biochemistry, Centre for Human and Molecular Biology (ZHMB), Saarland University, D-66123 Saarbrücken, Germany. [3]These authors contributed equally: Fabian Geissel, Lukas Lang. ✉e-mail: deponte@chemie.uni-kl.de

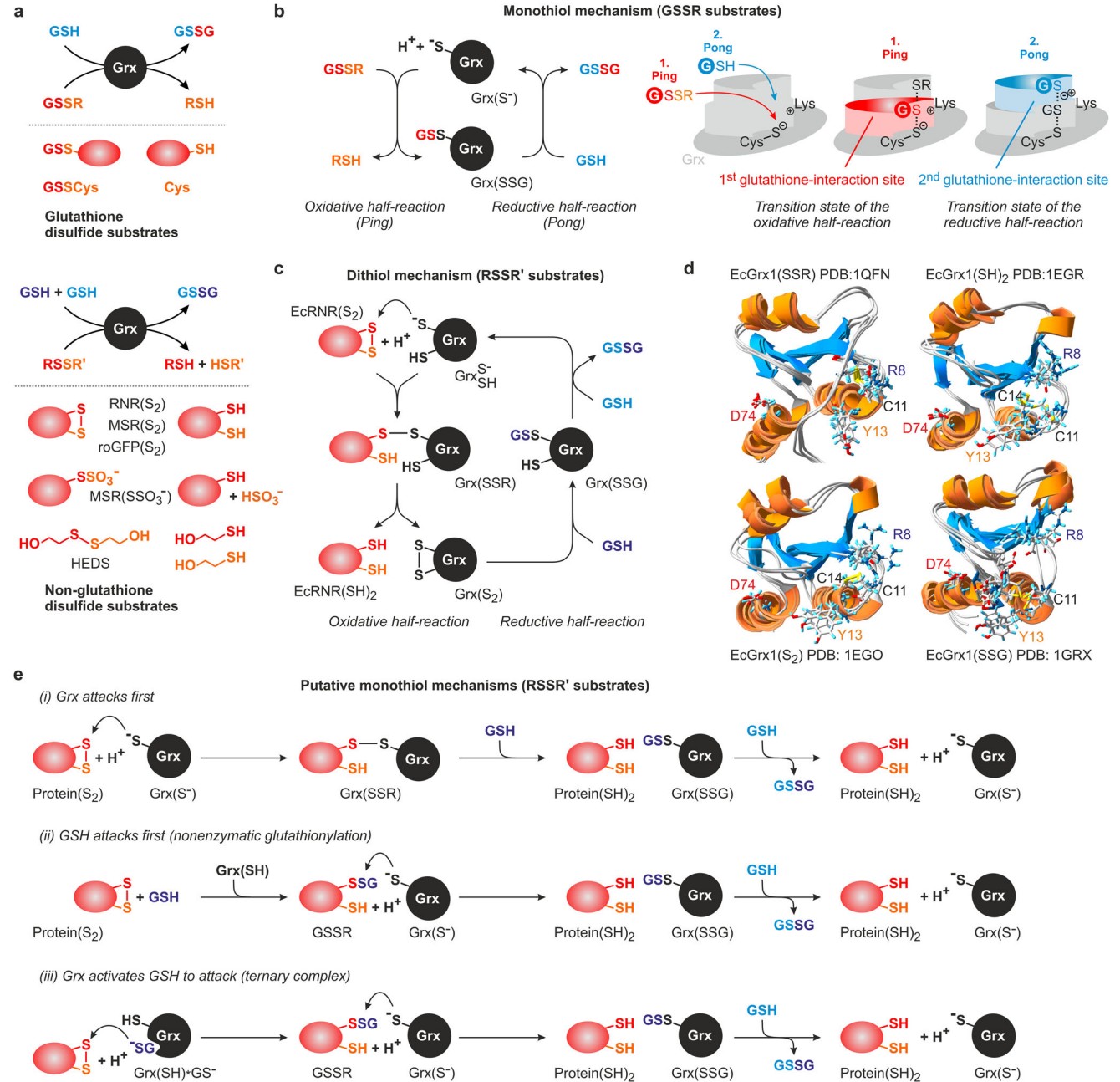

**Fig. 1 | Overview of the substrates and catalytic mechanisms of class I glutaredoxins. a** Glutaredoxins reduce glutathione- and non-glutathione disulfide substrates using one or two molecules of GSH (shown at the top and bottom, respectively). **b** Glutaredoxins use a monothiol ping-pong mechanism for the reduction of glutathione disulfide substrates. The glutathione moieties of GSSR and GSH are not identical and have to interact with different protein areas of the reduced and glutathionylated enzyme. **c** The reduction of selected non-glutathione disulfide substrates via a dithiol mechanism requires a second active-site cysteinyl residue. **d** NMR-structures of the mixed disulfide between EcGrx1 and a peptide of EcRNR, EcGrx1 disulfide, glutathionylated EcGrx1, and reduced EcGrx1 in accordance with the dithiol mechanism (depicted counterclockwise). The second cysteinyl residue was removed to stabilize the mixed disulfide species. Selected residues that were previously shown to interact with glutathione or to affect catalysis in related enzymes are highlighted. **e** Three alternative monothiol mechanisms for the reduction of non-glutathione disulfide substrates have been suggested (see text for details). All shown reactions are reversible.

mechanistic insights were gained to decipher the glutaredoxin-catalyzed reduction of glutathione disulfide substrates, there are still major gaps in our understanding of the enzyme mechanism regarding the reduction of non-glutathione disulfide substrates.

Many class I glutaredoxins are so-called dithiol enzymes because they have two cysteinyl residues in a C[P/S]Y[C/S] motif at their active site. The reversible reduction of GSSR substrates usually requires only the first active-site cysteinyl residue and is therefore catalyzed by a monothiol mechanism with ping-pong bi-bi kinetic patterns (Fig. 1b):[7–12,15,17,19,30] During the oxidative half-reaction the thiolate group

of the reduced glutaredoxin attacks the GSSR disulfide bond, yielding the first product (RSH) and the glutathionylated glutaredoxin Grx(SSG) as an intermediate. The Grx(SSG) intermediate is then reduced by GSH during the reductive half-reaction, yielding the reduced enzyme and GSSG. The glutathione moieties of GSSR and GSH have to interact with different protein areas as a result of the $S_N2$ geometry of both half-reactions of the ping pong bi-bi mechanism (Fig. 1b)[7,13–15]. The 1st glutathione-interaction site, also termed glutathione-scaffold site, is rather well defined and comprises several protein areas that altogether contribute to catalysis by interacting with

the glutathione moiety of GSSR[7,8,13–15,35], therefore explaining the specificity for the γ-glutamyl moiety[9,49,50]. This interaction site is also occupied by the GS⁻ ligand of iron-sulfur clusters that are bound to class II glutaredoxins[7,15,33,51–53]. The 2nd glutathione-interaction site appears to preferentially interact with the γ-glutamylcysteinyl moiety of GSH and is thought to facilitate the recruitment and activation of GSH as a preferred reducing agent[7,11,13,17,50]. This interaction site is less defined but comprises the glutathione moiety of Grx(SSG), parts of helix 3, and a conserved lysine/arginine residue that also contributes to the scaffold site[14,15]. Coupled steady-state kinetic measurements with GSSR and GSH revealed that both half-reactions usually occur with second order rate constants around $10^5$ to $10^6$ $M^{-1}$ $s^{-1}$ with the reductive half-reaction being slower for wild-type glutaredoxins[11,14,15,17,19].

The reversible reduction of $RS_2$ and RSSR' substrates can be catalyzed by a monothiol and/or dithiol mechanism depending on the substrate and organism[5–8]. Numerous artificial and natural monothiol class I glutaredoxins efficiently reduce HEDS or redox-sensitive fluorescent proteins without a second active-site cysteinyl residue[12,15,18,26,30,31,33–35,54–56]. Furthermore, monothiol class I glutaredoxins can catalyze all essential glutaredoxin- and thioredoxin-dependent protein disulfide reductions in yeast, including, for example, the reduction of RNR and MSR[34]. A monothiol mechanism was also reported for the in vitro reduction of mammalian RNR[46] and can be also employed, although with a decreased activity, by a Grx-RNR fusion protein from *Facklamia ignava*[47]. In contrast, the in vitro reduction of *Escherichia coli* RNR or PR was shown to require both active-site cysteinyl residues of glutaredoxin 1 (EcGrx1)[41,57]. The suggested dithiol mechanism for the reduction of EcRNR by EcGrx1 comprises a dithiol-disulfide exchange reaction followed by the stepwise reduction of EcGrx1($S_2$) and EcGrx1(SSG) by two molecules of GSH (Fig. 1c, d)[57,58]. According to the Cleland nomenclature the reaction should follow a uni-uni-bi-uni ping-pong mechanism[8,59], although detailed kinetic studies to confirm the mechanism for EcGrx1 or the *F. ignava* fusion enzyme are still missing. It is plausible to postulate that the first and second GSH molecule should interact during the reductive half-reaction of the dithiol mechanism with the 1st and 2nd glutathione-interaction site, respectively. The monothiol mechanism for the reduction of non-glutathione disulfides is controversial and at least three different nonexclusive mechanisms have been discussed to date (Fig. 1e)[3,9,14,15,30,34,46,57,60]: (i) The thiolate of the reduced monothiol class I glutaredoxin directly attacks the disulfide substrate followed by the stepwise reduction of the mixed disulfide and Grx(SSG) by two molecules of GSH. There are two variations of the suggested mechanism depending on whether the first GSH molecule attacks the Grx(SSR) intermediate at the sulfur atom of the glutaredoxin[60] or the substrate[15,30]. (ii) GSH nonenzymatically reacts with the disulfide substrate, yielding a GSSR substrate that is reduced as described above[9,46,57]. (iii) GSH is activated by the glutaredoxin and catalysis requires a ternary complex with the disulfide substrate[30,34]. Following the enzyme-catalyzed glutathionylation of the substrate, the deglutathionylation occurs as decribed above. Since all reactions in Fig. 1 are reversible, class I glutaredoxins can also catalyze the GSSR- or GSSG-dependent glutathionylation of thiols or the formation of (protein) disulfides[3,5,35,61]. Such oxidations are in direct competition with glutathione reductase and ABC transporters which usually maintain very low GSSG concentrations under physiological conditions[5,7,62–64].

Most of the commonly used oxidoreductase assays for glutaredoxins are either discontinuous and labor-intensive or unidirectional and coupled to the glutathione reductase-dependent consumption of NADPH[1,2,6,8,16–18,30]. One exception is the analysis of fluorescent protein substrates such as rxYFP or roGFP2, which allow a direct continuous detection of the reversible glutaredoxin-dependent protein disulfide reduction and formation in vitro[14,15,31–35]. Genetically encoded rxYFP or roGFP2 probes have also become extremely valuable tools for noninvasive, ratiometric real-time redox measurements in various compartments, cells, and organisms[64–71]. Fusion constructs between glutaredoxins or peroxidases and roGFP2 yield specific redox sensors for monitoring spatiotemporal changes of intracellular glutathione or peroxide concentrations[32,61,62,69,72–74]. Furthermore, roGFP2 fusion constructs with wild-type and mutant enzymes allow the intracellular analysis of enzyme mechanisms and structure-function relationships[15,34,35,75]. A current drawback of intracellular roGFP2-based mechanistic studies is, however, that the mechanisms and structure-function relationships still have to be validated in vitro. Deciphering the reversible glutaredoxin-dependent redox reactions of roGFP2 therefore not only closes a gap in our knowledge of glutaredoxin catalysis but also facilitates the intrepretation of roGFP2 responses from redox measurements and mechanistic analyses in vivo.

Here we determined the rate constants for the individual redox reactions between the two redox-sensitive cysteinyl residues of roGFP2 and the active-site cysteinyl residues of the model class I glutaredoxin PfGrx from the malaria parasite *Plasmodium falciparum* using stopped-flow kinetic measurements in combination with redox mobility shift assays. PfGrx shares 39% sequence identity with human Grx1, including a CPYC motif and a semiconserved GGC motif, and is well characterized in vitro and in roGFP2 assays in yeast[14,26,30,35]. We show (i) that PfGrx-catalyzed redox reactions outcompete uncatalyzed reactions between roGFP2 and GSH or GSSG by several orders of magnitude, (ii) that the PfGrx-catalyzed reduction and oxidation of roGFP2 occur via a mono- and not a dithiol mechanism, and (iii) that the reduction of roGFP2($S_2$) as a model non-glutathione disulfide substrate most likely involves a rate-limiting glutathionylation in a ternary complex with a GSH molecule that is activated at the 1st glutathione-interaction site of PfGrx. Furthermore, we confirm that the initial glutathionylation of the glutaredoxin moiety and not the glutathione transfer to the roGFP2 moiety is rate-limiting for the oxidation of fusion constructs at physiological or low micromolar GSSG concentrations, thus allowing glutaredoxin structure-function analysis in vivo.

## Results

### PfGrx(SSG)-dependent oxidation of reduced wild-type and monothiol roGFP2

GSSG rapidly oxidizes roGFP2 in the presence of glutaredoxins in vitro and in vivo[33–35,61,62]. To analyze the glutaredoxin-dependent oxidation kinetics for recombinant roGFP2, we first confirmed the fluorescence spectra of reduced and oxidized wild-type roGFP2 (roGFP2$^{WT}$(SH)$_2$ and roGFP2$^{WT}$($S_2$), respectively) using a stopped-flow spectrofluorometer (Fig. 2a). Oxidation of roGFP2$^{WT}$ resulted in an increased fluorescence at an excitation of 400 nm and a decreased fluorescence at an excitation of 484 nm as reported previously[66]. To address the reactivities of the redox-sensitive cysteinyl residues, we also analyzed the recombinant monothiol roGFP2 variants roGFP2$^{C151S}$ and roGFP2$^{C208S}$ and detected small but reproducible glutathionylation-dependent changes of fluorescence at both excitation wavelengths for roGFP2$^{C151S}$(SSG) and roGFP2$^{C208S}$(SSG) (Fig. 2a). While glutathionylation of roGFP2$^{C208S}$ also caused an increased fluorescence at an excitation of 400 nm and a decreased fluorescence at an excitation of 484 nm, glutathionylation of roGFP2$^{C151S}$ resulted in a decreased fluorescence at both excitation wavelengths. To exclude that the small spectral changes reflected just a partial glutathionylation, we performed electrophoretic mobility shift assays (Fig. S1a). These assays confirmed that residue C208 in roGFP2$^{C151S}$ and residue C151 in roGFP2$^{C208S}$ were protected from alkylation in accordance with a complete glutathionylation under the chosen conditions.

The spectral changes at both excitation wavelengths followed identical kinetics, which allowed us to reliably monitor the formation of roGFP2$^{WT}$($S_2$), roGFP2$^{C151S}$(SSG), or roGFP2$^{C208S}$(SSG) using PfGrx$^{C32S/C88S}$(SSG) as an oxidant (Fig. 2b). PfGrx$^{C32S/C88S}$ is the previously characterized monothiol mutant of PfGrx. C32 is the second active-site cysteinyl residue of PfGrx and C88 is the third cysteinyl residue, which

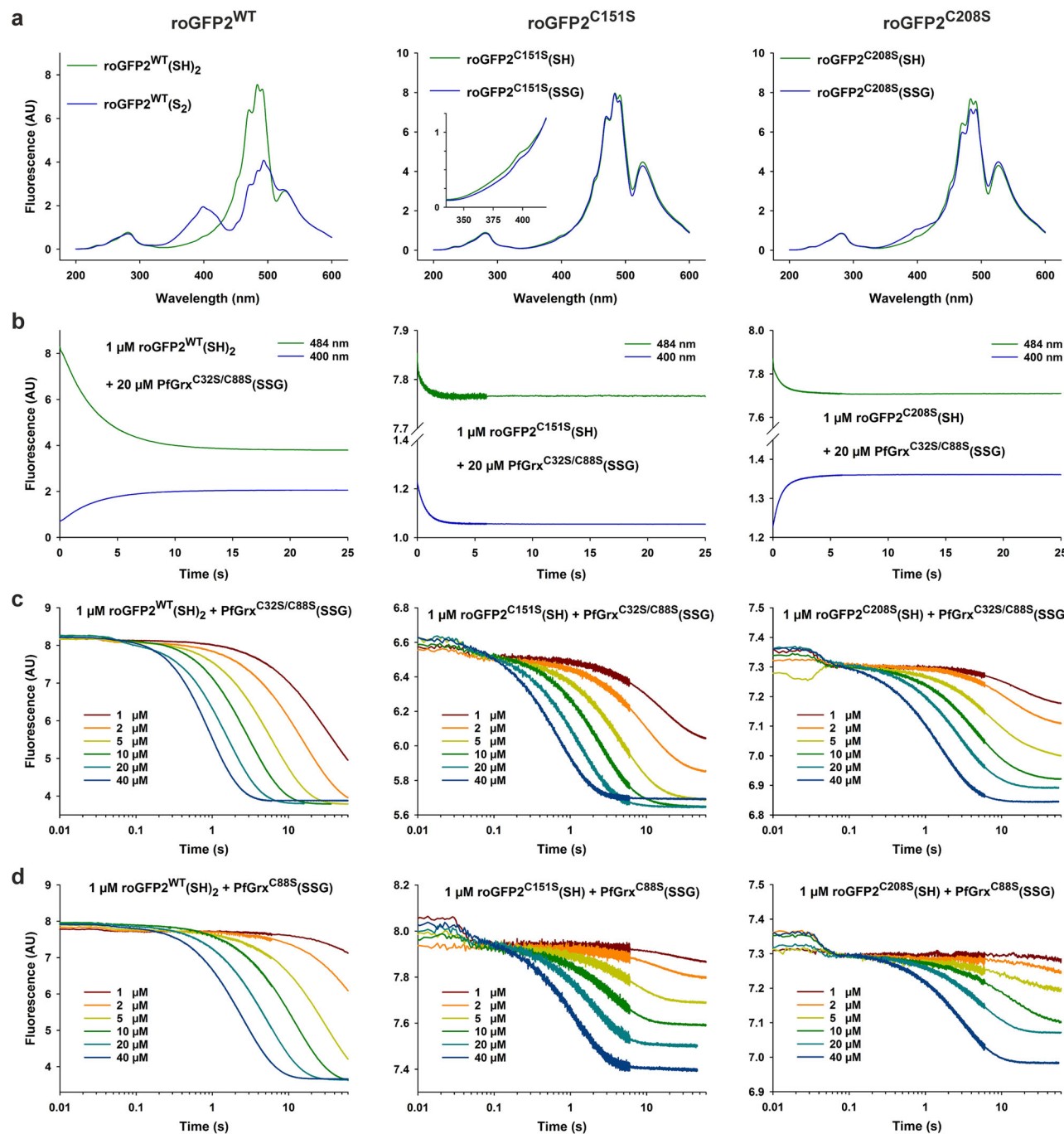

**Fig. 2 | Spectra and oxidation kinetics for reduced roGFP2^WT, roGFP2^C151S, and roGFP2^C208S. a** Fluorescence spectra of 1 μM roGFP2^WT(SH)_2 and roGFP2^WT(S_2) are shown on the left. The spectrum for roGFP2^WT(S_2) was recorded 2 min after the addition of 20 μM PfGrx^C32S/C88S(SSG). The fluorescence spectra of 1 μM roGFP2^C151S(SH) and roGFP2^C208S(SH) are shown in the middle and on the right, respectively. Both monothiol roGFP2 variants were incubated with 10 mM GSSG to obtain the spectra of roGFP2^C151S(SSG) and roGFP2^C208S(SSG), which were recorded after the removal of GSSG and GSH. **b** Representative stopped-flow oxidation kinetics for the reaction between 1 μM reduced wild-type or mutant roGFP2 with 20 μM PfGrx^C32S/C88S(SSG). **c, d** Representative stopped-flow oxidation kinetics at 484 nm for the reaction between 1 μM reduced wild-type or mutant roGFP2 with the indicated concentrations of either PfGrx^C32S/C88S(SSG) or PfGrx^C88S(SSG).

is in the proximity of the active site and can also undergo redox modifications[14,26,30]. The PfGrx^C32S/C88S(SSG)-dependent oxidation of roGFP2^WT or glutathionylation of both monothiol roGFP2 variants were accelerated at increasing glutaredoxin concentrations and followed pseudo-first-order kinetics (Fig. 2c). Furthermore, the endpoint fluorescence indicated quite different redox potentials for the roGFP2 variants. While roGFP2^WT was completely oxidized at even very low PfGrx^C32S/C88S(SSG) concentrations, 1 μM PfGrx^C32S/C88S(SSG) glutathionylated approximately 50% of 1 μM roGFP2^C151S(SH) suggesting that the redox couples roGFP2^C151S(SSG)/roGFP2^C151S(SH) and PfGrx^C32S/C88S(SSG)/PfGrx^C32S/C88S(SH) have very similar redox potentials. In contrast to roGFP2^C151S(SH), 50% glutathionylation of 1 μM roGFP2^C208S(SH) required almost 5 μM PfGrx^C32S/C88S(SSG) (Fig. 2c). The results and data interpretations of the endpoint fluorescences were independently validated by electrophoretic mobility shift assays (Fig. S1b). Pseudo-first-order kinetics were also detected using PfGrx^C88S(SSG), the freshly glutathionylated dithiol variant of PfGrx[26], as an oxidant (Fig. 2d). In these assays, roGFP2^WT(SH)_2 was again more reducing than

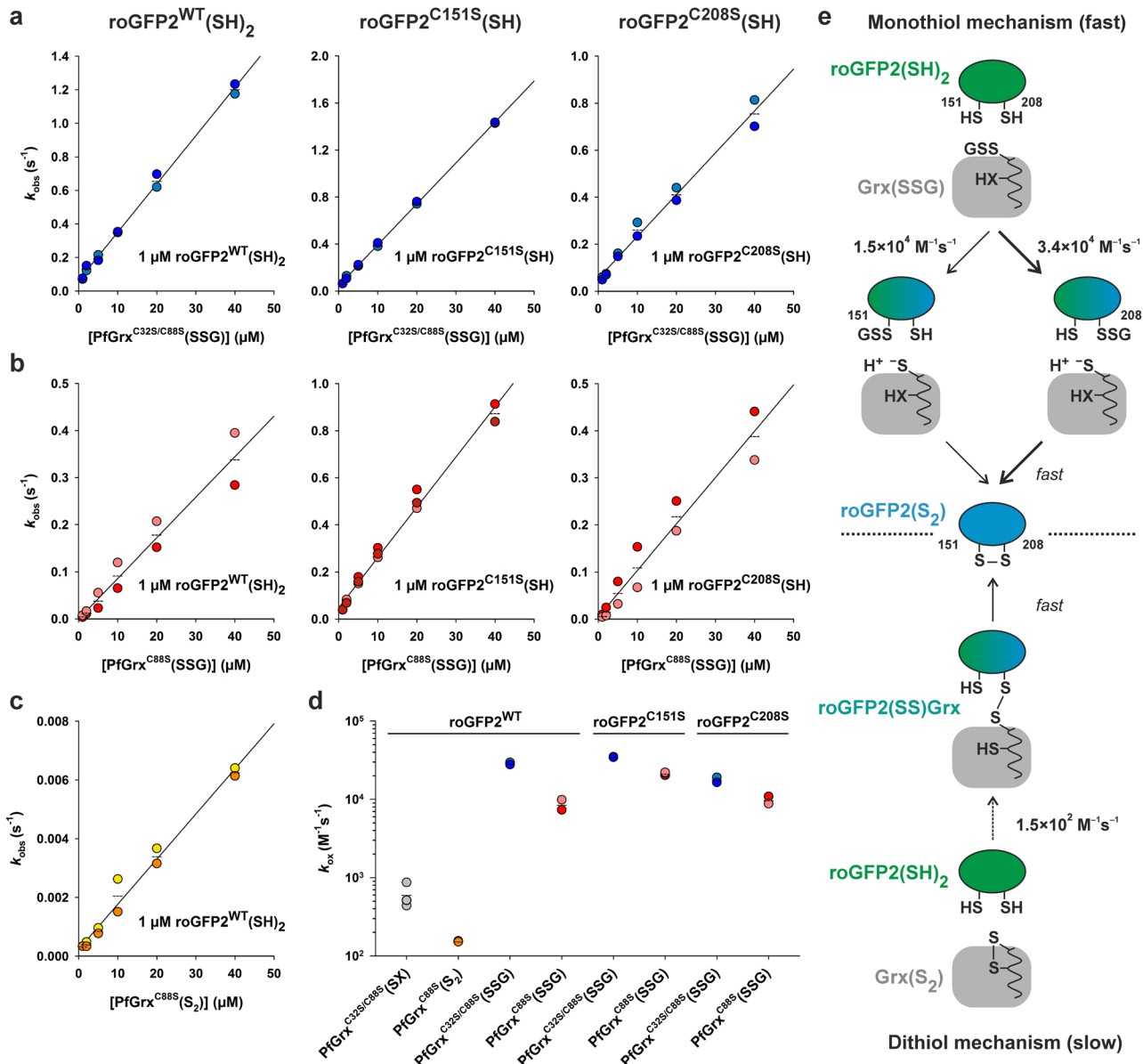

**Fig. 3 | Rate constants for the oxidation of wild-type and monothiol roGFP2.** Secondary plots for the reaction kinetics of 1 μM reduced wild-type or monothiol roGFP2 with (**a**) PfGrx$^{C32S/C88S}$(SSG), (**b**) PfGrx$^{C88S}$(SSG), or (**c**) PfGrx$^{C88S}$(S$_2$). **d** Overview of the second order rate constants $k_{ox}$ for the oxidation of the indicated roGFP2 variants by the indicated PfGrx mutants. The rate constants are also listed in Table 1. **e** Data interpretation of the roGFP2 oxidation kinetics. The preferred oxidation pathway is highlighted with thick arrows. All data sets were generated from at least two independent biological replicates.

roGFP2$^{C151S}$(SH), which was again more reducing than roGFP2$^{C208S}$(SH). A comparison of the endpoint fluorescences also revealed that about five times higher PfGrx$^{C88S}$(SSG) than PfGrx$^{C32S/C88S}$(SSG) concentrations were necessary to obtain a 50% oxidation of the roGFP2 variants (Fig. 2c, d). The results and data interpretations were again confirmed by electrophoretic mobility shift assays (Fig. S1b).

In summary, we established stopped-flow kinetic assays for the PfGrx-dependent oxidation of roGFP2$^{WT}$ and glutathionylation of its monothiol variants and estimated the following order of redox potentials for roGFP2$^{C208S}$(SSG)/roGFP2$^{C208S}$(SH) > roGFP2$^{C151S}$(SSG)/roGFP2$^{C151S}$(SH) ≈ PfGrx$^{C32S/C88S}$(SSG)/PfGrx$^{C32S/C88S}$(SH) > PfGrx$^{C88S}$(SSG)/PfGrx$^{C88S}$(SH) > roGFP2$^{WT}$(S$_2$) /roGFP2$^{WT}$(SH)$_2$.

### Rate constants for the PfGrx-dependent oxidation of wild-type and monothiol roGFP2

Next, we fitted the pseudo-first-order glutathionylation kinetics at 484 nm from Fig. 2 and plotted the $k_{obs}$ values against the

PfGrx$^{C32S/C88S}$(SSG) or PfGrx$^{C88S}$(SSG) concentration to determine the second-order rate constants for the formation of roGFP2$^{WT}$(S$_2$), roGFP2$^{C151S}$(SSG), or roGFP2$^{C208S}$(SSG) from the slopes of the linear fits (Fig. 3, Table 1). Both monothiol roGFP2 variants were rapidly glutathionylated with residue C208 reacting about twice as fast as residue C151 regardless whether PfGrx$^{C32S/C88S}$(SSG) or PfGrx$^{C88S}$(SSG) was used as the oxidant. The hightest second-order rate constant $k_{ox}$ of 3.5 × 10$^4$ M$^{-1}$ s$^{-1}$ was determined for the reaction between roGFP2$^{C151S}$(SH) and PfGrx$^{C32S/C88S}$(SSG) (Fig. 3a–d). A similar overall rate constant of 3.0 × 10$^4$ M$^{-1}$ s$^{-1}$ was obtained for the PfGrx$^{C32S/C88S}$(SSG)-dependent formation of roGFP2$^{WT}$(S$_2$). Thus, glutathionylation of roGFP2$^{WT}$(SH)$_2$ was most likely rate-limiting before the intramolecular disulfide bond was rapidly formed. PfGrx$^{C32S/C88S}$(SSG) catalyzed the oxidation of roGFP2 via a monothiol mechanism and reacted with all roGFP2 variants 1.5 to 3 times faster than PfGrx$^{C88S}$(SSG) (Fig. 3a, b, Table 1).

We also analyzed the oxidation kinetics for roGFP2$^{WT}$ using the mono- and dithiol variants PfGrx$^{C32S/C88S}$ and PfGrx$^{C88S}$ following pre-

**Table 1 | Rate constants for the redox reaction of PfGrx, roGFP2, and related compounds**

| Assigned reaction partners A/B/C and products P/Q | | | | Rate constant | pH/temp. |
|---|---|---|---|---|---|
| A | + B ( + C) | P | Q | | |
| GSH | roGFP2$^{WT}$(S$_2$) | | roGFP2$^{WT}$(SSG) | Not detectable | 8.0/25 °C |
| PfGrx$^{C88S}$(SH)$_2$*roGFP2(S$_2$) | GSH | PfGrx$^{C88S}$(SH)$_2$ | roGFP2$^{WT}$(SSG) | ca. 10 M$^{-1}$ s$^{-1}$ [a] | 8.0/25 °C |
| PfGrx$^{C88S}$(SH)$_2$*GSH | roGFP2$^{WT}$(S$_2$) | PfGrx$^{C88S}$(SH)$_2$ | roGFP2$^{WT}$(SSG) | ca. $10^3$ M$^{-1}$ s$^{-1}$ [a] | 8.0/25 °C |
| PfGrx$^{C88S}$(SH)$_2$ | roGFP2$^{WT}$(S$_2$) + GSH | PfGrx$^{C88S}$(SH)$_2$ | roGFP2$^{WT}$(SSG) | $7.3 \times 10^5$ M$^{-2}$ s$^{-1}$ | 8.0/25 °C |
| PfGrx$^{DM}$(SH) [b] | roGFP2$^{WT}$(S$_2$) + GSH | PfGrx$^{DM}$(SH) | roGFP2$^{WT}$(SSG) | $1.3 \times 10^5$ M$^{-2}$ s$^{-1}$ | 8.0/25 °C |
| PfGrx$^{C88S}$(SH)$_2$ | roGFP2$^{WT}$(S$_2$) + GSH | PfGrx$^{C88S}$(SH)$_2$ | roGFP2$^{WT}$(SSG) | $3.4 \times 10^3$ M$^{-1}$ s$^{-1}$ [c] | 8.0/25 °C |
| PfGrx$^{DM}$(SH) [b] | roGFP2$^{WT}$(S$_2$) + GSH | PfGrx$^{DM}$(SH) | roGFP2$^{WT}$(SSG) | $6.5 \times 10^2$ M$^{-1}$ s$^{-1}$ [c] | 8.0/25 °C |
| PfGrx$^{K26A/DM}$(SH) [b] | roGFP2$^{WT}$(S$_2$) + GSH | PfGrx$^{K26A/DM}$(SH) | roGFP2$^{WT}$(SSG) | 36 M$^{-1}$ s$^{-1}$ [c] | 8.0/25 °C |
| PfGrx$^{D90A/DM}$(SH) [b] | roGFP2$^{WT}$(S$_2$) + GSH | PfGrx$^{D90A/DM}$(SH) | roGFP2$^{WT}$(SSG) | $5.1 \times 10^2$ M$^{-1}$ s$^{-1}$ [c] | 8.0/25 °C |
| PfGrx$^{DM}$(SH) [b] | GSSCys | PfGrx$^{DM}$(SSG) | Cys | $1.3 \times 10^6$ M$^{-1}$ s$^{-1}$ [d] | 8.0/25 °C |
| PfGrx$^{K26A/DM}$(SH) [b] | GSSCys | PfGrx$^{K28A/DM}$(SSG) | Cys | $6.9 \times 10^4$ M$^{-1}$ s$^{-1}$ [d] | 8.0/25 °C |
| PfGrx$^{D90A/DM}$(SH) [b] | GSSCys | PfGrx$^{D90A/DM}$(SSG) | Cys | $1.0 \times 10^6$ M$^{-1}$ s$^{-1}$ [d] | 8.0/25 °C |
| PfGrx$^{C88S}$(SH)$_2$ | roGFP2$^{C151S}$(SSG) | PfGrx$^{C88S}$(SSG) | roGFP2$^{C151S}$(SH) | $4.9 \times 10^5$ M$^{-1}$ s$^{-1}$ | 8.0/25 °C |
| PfGrx$^{DM}$(SH) [b] | roGFP2$^{C151S}$(SSG) | PfGrx$^{DM}$(SSG) | roGFP2$^{C151S}$(SH) | $6.4 \times 10^4$ M$^{-1}$ s$^{-1}$ | 8.0/25 °C |
| PfGrx$^{C88S}$(SH)$_2$ | roGFP2$^{C208S}$(SSG) | PfGrx$^{DM}$(SSG) | roGFP2$^{C208S}$(SH) | $1.4 \times 10^6$ M$^{-1}$ s$^{-1}$ | 8.0/25 °C |
| PfGrx$^{DM}$(SH) [b] | roGFP2$^{C208S}$(SSG) | PfGrx$^{DM}$(SSG) | roGFP2$^{C208S}$(SH) | $2.5 \times 10^5$ M$^{-1}$ s$^{-1}$ | 8.0/25 °C |
| PfGrx$^{DM}$(SH) [b] | roGFP2$^{WT}$(SSG) | PfGrx$^{DM}$(SSG) | roGFP2$^{WT}$(SH)$_2$ | ca. $8 \times 10^4$ M$^{-1}$ s$^{-1}$ [a] | 8.0/25 °C |
| rxYFP(SH)$_2$-ScGrx1$^{WT}$(SH)$_2$ | GSSG | rxYFP-ScGrx1$^{WT}$(SSG) | GSH | $4 \times 10^3$ M$^{-1}$ s$^{-1}$ [e] | 7.0/30 °C |
| rxYFP(SH)$_2$-ScGrx1$^{CxxS}$(SH) | GSSG | rxYFP-ScGrx1$^{CxxS}$(SSG) | GSH | $3 \times 10^4$ M$^{-1}$ s$^{-1}$ [e] | 7.0/30 °C |
| roGFP2(SH)$_2$-PfGrx$^{WT}$(SH)$_2$ | GSSG | roGFP2-PfGrx$^{WT}$(SSG) | GSH | $4.3 \times 10^7$ M$^{-1}$ s$^{-1}$ | 8.0/25 °C |
| roGFP2(SH)$_2$-PfGrx$^{C88S}$(SH)$_2$ | GSSG | roGFP2-PfGrx$^{C88S}$(SSG) | GSH | $1.4 \times 10^7$ M$^{-1}$ s$^{-1}$ | 8.0/25 °C |
| roGFP2(SH)$_2$-PfGrx$^{DM}$(SH) [b] | GSSG | roGFP2-PfGrx$^{DM}$(SSG) | GSH | $2.6 \times 10^6$ M$^{-1}$ s$^{-1}$ | 8.0/25 °C |
| roGFP2$^{WT}$(SH)$_2$-PfGrx$^{WT}$(SSG) | | roGFP2(SSG)-PfGrx$^{WT}$(SH)$_2$ | | 9.7 s$^{-1}$ | 8.0/25 °C |
| roGFP2$^{WT}$(SH)$_2$-PfGrx$^{C88S}$(SSG) | | roGFP2(SSG)-PfGrx$^{C88S}$(SH)$_2$ | | 11.6 s$^{-1}$ | 8.0/25 °C |
| roGFP2$^{WT}$(SH)$_2$-PfGrx$^{DM}$(SSG) [b] | | roGFP2(SSG)-PfGrx$^{DM}$(SH) | | 11.5 s$^{-1}$ | 8.0/25 °C |
| GSSG | rxYFP(SH)$_2$ | rxYFP(SSG) | GSH | 1.2 M$^{-1}$ s$^{-1}$ [f] | 7.0/30 °C |
| GSSG | roGFP2$^{WT}$(SH)$_2$ | roGFP2$^{WT}$(SSG) | GSH | 0.7 M$^{-1}$ s$^{-1}$ | 8.0/25 °C |
| PfGrx$^{C88S}$(S$_2$) | roGFP2$^{WT}$(SH)$_2$ | PfGrx$^{C88S}$(SS)roGFP2$^{WT}$ | | $1.5 \times 10^2$ M$^{-1}$ s$^{-1}$ | 8.0/25 °C |
| PfGrx$^{C88S}$(SSG) | roGFP2$^{WT}$(SH)$_2$ | PfGrx$^{C88S}$(SH)$_2$ | roGFP2$^{WT}$(SSG) | $8.6 \times 10^3$ M$^{-1}$ s$^{-1}$ | 8.0/25 °C |
| PfGrx$^{DM}$(SSG) [b] | roGFP2$^{WT}$(SH)$_2$ | PfGrx$^{DM}$(SH) | roGFP2$^{WT}$(SSG) | $3.0 \times 10^4$ M$^{-1}$ s$^{-1}$ | 8.0/25 °C |
| PfGrx$^{C88S}$(SSG) | roGFP2$^{C151S}$(SH) | PfGrx$^{C88S}$(SH)$_2$ | roGFP2$^{C151S}$(SSG) | $2.1 \times 10^4$ M$^{-1}$ s$^{-1}$ | 8.0/25 °C |
| PfGrx$^{DM}$(SSG) [b] | roGFP2$^{C151S}$(SH) | PfGrx$^{DM}$(SH) | roGFP2$^{C151S}$(SSG) | $3.5 \times 10^4$ M$^{-1}$ s$^{-1}$ | 8.0/25 °C |
| PfGrx$^{C88S}$(SSG) | roGFP2$^{C208S}$(SH) | PfGrx$^{C88S}$(SH)$_2$ | roGFP2$^{C208S}$(SSG) | $9.9 \times 10^3$ M$^{-1}$ s$^{-1}$ | 8.0/25 °C |
| PfGrx$^{DM}$(SSG) [b] | roGFP2$^{C208S}$(SH) | PfGrx$^{DM}$(SH) | roGFP2$^{C208S}$(SSG) | $1.8 \times 10^4$ M$^{-1}$ s$^{-1}$ | 8.0/25 °C |
| PfGrx$^{DM}$(SSG) [b] | GSH | PfGrx$^{DM}$(SH) | GSSG | $1.2 \times 10^5$ M$^{-1}$ s$^{-1}$ [d] | 8.0/25 °C |
| PfGrx$^{K28A/DM}$(SSG) [b] | GSH | PfGrx$^{K26A/DM}$(SH) | GSSG | $5.0 \times 10^4$ M$^{-1}$ s$^{-1}$ [d] | 8.0/25 °C |
| PfGrx$^{D90A/DM}$(SSG) [b] | GSH | PfGrx$^{D90A/DM}$(SH) | GSSG | $1.7 \times 10^5$ M$^{-1}$ s$^{-1}$ [d] | 8.0/25 °C |

[a] Values from simulations (Fig. 7);
[b] DM = C32S/C88S;
[c] $k_{red}^{app}$ values at 5 mM GSH;
[d] From ref. 14.
[e] From ref. 31.
[f] From ref. 67.

treatment with the disulfide bond-inducing redox reagent 3-(dimethylcarbamoylimino)−1,1-dimethylurea (Diamide)[76] (Fig. 3c, d, Table 1). Diamide-treated monothiol PfGrx$^{C32S/C88S}$ was intended as a negative control but had a detectable activity with roGFP2$^{WT}$(SH)$_2$ with a rate constant $k_{ox}$ around $6.5 \times 10^2$ M$^{-1}$ s$^{-1}$ (Fig. 3d). The unexpected activity can be explained by the formation of variable amounts of disulfide-bridged PfGrx$^{C32S/C88S}$ homodimers and potential Diamide-modified monomers, summarized as PfGrx$^{C32S/C88S}$(SX), as revealed by non-reducing SDS-PAGE (Fig. S1c). PfGrx$^{C88S}$(S$_2$) oxidized roGFP2$^{WT}$(SH)$_2$ only slowly with a second order rate of $1.5 \times 10^2$ M$^{-1}$ s$^{-1}$ (Fig. 3c, d). Thus, roGFP2 oxidation by PfGrx occurs preferentially via a glutathione-dependent monothiol and not a glutathione-independent dithiol mechanism (Fig. 3d, e). The slow reaction between PfGrx$^{C88S}$(S$_2$) and roGFP2$^{WT}$(SH)$_2$ was confirmed by electrophoretic mobility shift assays (Fig. S1d). No mixed disulfide intermediate was detected in accordance with a slow reaction between both proteins that was followed by a rapid intramolecular disulfide bond formation yielding roGFP2$^{WT}$(S$_2$). The instability of a mixed disulfide between roGFP2 and PfGrx was also supported by unsuccessful attempts to force and stabilize a Grx(SSR) species in titration experiments with roGFP2$^{C151S}$(SH) and up to 80 μM PfGrx$^{C88S}$(S$_2$) (Fig. S1e). The second order rate constant $k_{ox}$ for the nonenzymatic reaction between GSSG and roGFP2$^{WT}$(SH)$_2$ was 0.7 M$^{-1}$ s$^{-1}$ (Fig. S2), clearly demonstrating that uncatalyzed glutathionylation of regular protein thiols occurs much

too slowly at low physiological GSSG concentrations to compete with glutaredoxin-dependent glutathionylation[63].

In summary, residues C151 and C208 of reduced roGFP2 are both susceptible to enzyme-catalyzed glutathionylation with residue C208 reacting twice as fast as residue C151. Mutation of residue C32 of PfGrx results in an up to three times faster glutathionylation of roGFP2 with a $k_{ox}$ of $3.5 \times 10^4\,M^{-1}\,s^{-1}$. Furthermore, the PfGrx-catalyzed oxidation of roGFP2 is more than four orders of magnitude faster than the uncatalyzed reaction with GSSG and occurs two orders of magnitude faster via the glutathione-dependent monothiol mechanism than by the glutathione-independent dithiol mechanism (Fig. 3e, Table 1). Thus, roGFP2 oxidation does not efficiently occur via a Grx(SSR) intermediate and we can exclude the dithiol mechanism and monothiol mechanism (i) for roGFP2 oxidation (reverse reactions in Fig. 1c, e). Taking into account the $S_N2$ reaction geometry, the size of roGFP2, and the accessibility of both sulfur atoms in Grx(SSG) (Fig. 1b), we favor an attack of roGFP2 at the glutathione sulfur atom as the most likely roGFP2 oxidation pathway (reverse reaction from mechanism (ii) or (iii) from Fig. 1e).

## Oxidation kinetics for roGFP2$^{WT}$-PfGrx fusion constructs

The in vivo oxidation kinetics from fusion constructs between roGFP2$^{WT}$ and a variety of glutaredoxins and mutants thereof were shown to correlate very well with the rate constants for the oxidative half-reaction from GSSCys assays in vitro, suggesting that the glutathionylation of the glutaredoxin and not the transfer of oxidation to the fused roGFP2 moiety was rate-limiting under the chosen conditions[15,34,35]. To test this data interpretation, we analyzed the GSSG-dependent oxidation of reduced fusion constructs between roGFP2$^{WT}$ and wild-type PfGrx (PfGrx$^{WT}$), the dithiol mutant PfGrx$^{C88S}$, or the monothiol mutant PfGrx$^{C32S/C88S}$ (Fig. 4, Table 1). Two separate phases were observed at an excitation of 484 nm, a rapid but small increase and a slower but more pronounced decrease of fluorescence (Fig. 4a). Similar biphasic kinetics, although a 100-fold slower, were reported for the GSSG-dependent oxidation of fusion constructs between rxYFP and a mono- or dithiol variant of ScGrx1 from yeast[31]. Secondary plots of the $k_{obs}$ values from single exponential fits of the first phase and double exponential fits of the second phase against the GSSG concentration yielded a linear and hyperbolic correlation for the first and second phase, respectively (Fig. 4b–d). The second order rate constant for the first phase depended on the PfGrx variant and was as high as $4.3 \times 10^7\,M^{-1}\,s^{-1}$ for roGFP2$^{WT}$-PfGrx$^{WT}$. The rate constant for the fusion construct with dithiol PfGrx$^{C88S}$ was about three times smaller and decreased again fivefold for monothiol PfGrx$^{C32S/C88S}$ (Fig. 4b–e). This trend is in contrast to the rxYFP fusion constructs with mono- and dithiol variants of ScGrx1, which were shown to react more rapidly with GSSG when the second cysteinyl residue was replaced by serine (Table 1)[31]. We interpret the first phase as the glutathionylation of the PfGrx moiety that became detectable for the nearby fraction of fused roGFP2 molecules (Fig. 4f). Thus, glutathionylation of roGFP2$^{WT}$-PfGrx$^{C32S/C88S}$ by GSSG was about 2.5-times faster as the previously determined second-order rate constant for the glutathionylation of PfGrx$^{C32S/C88S}$ by GSSCys (Table 1)[14]. The faster reaction with GSSG might reflect the simultaneous interaction of the substrate with both glutathione interaction sites. An alternative interpretation for the first phase could be a rapid noncovalent interaction between PfGrx and GSSG in the vicinity of roGFP2, however, this seems less likely considering the smaller values for the structurally similar PfGrx mutants or, in particular, the rxYFP fusion constructs with ScGrx1 (Table 1). The secondary plot for the first $k_{obs}$ values from the second phase yielded a rate contant at GSSG saturation around $10\,s^{-1}$ for all three PfGrx variants (Fig. 4e). We assign this rate constant of the second phase to the intramolecular glutathione transfer within the roGFP2$^{WT}$-PfGrx fusion constructs. The strong decrease of fluorescence is then caused by the immediate formation of roGFP2$^{WT}$(S$_2$) which cannot be kinetically

separated from the slower glutathionylation of the roGFP2 moiety (Fig. 4f). One explanation for the necessity to use double exponential fits for the second phase could be that a fraction of the roGFP2$^{WT}$(SH)$_2$-PfGrx(SSG) intermediates glutathionylates the roGFP2 moiety of another molecule in trans.

In summary, the roGFP2 oxidation kinetics of PfGrx fusion constructs reflect at up to micromolar GSSG concentrations the glutathionylation kinetics of the glutaredoxin moiety in accordance with previous intracellular structure-function analyses and data interpretations[15,34,35]. The GSSG-dependent oxidation of roGFP2 by fused PfGrx variants is two to three orders of magnitude faster than the oxidation by unfused PfGrx(SSG) variants and more than seven orders of magnitude faster than the uncatalyzed reaction with GSSG (Table 1). The intramolecular glutathionylation of the roGFP2 moiety only becomes rate-limiting at saturating, medium micromolar GSSG concentrations. Such high GSSG concentrations are unlikely to occur in most subcellular compartments but could be reached in the secretory pathway or during titration experiments of transgenic cell lines with an upregulated GSSG transporter[35,63,64,77].

## PfGrx-dependent reduction kinetics for glutathionylated monothiol roGFP2

The glutaredoxin-dependent reduction of glutathionylated monothiol roGFP2 variants was analyzed in the presence and absence of GSH (Fig. 5 and Table 1). When GSH was tested together with reduced PfGrx$^{C88S}$ or PfGrx$^{C32S/C88S}$, the syringe of the stopped-flow spectrofluorometer also contained 0.4 U/mL glutathione reductase (GR) and 0.25 mM NADPH to ensure the removal of trace amounts of GSSG that might interfere with the redox measurements.

The PfGrx-dependent deglutathionylations of roGFP2$^{C151S}$(SSG) and roGFP2$^{C208S}$(SSG) both occurred rapidly in the absence of GSH (Fig. 5a, b). High GSH concentrations in the reaction mixtures even slightly decreased the rate constants, suggesting a competition for the 1st glutathione-interaction site (Fig. 1b). The highest second order rate constant $k_{red}$ of $1.4 \times 10^6\,M^{-1}\,s^{-1}$ was determined for the reaction between PfGrx$^{C88S}$(SH)$_2$ and roGFP2$^{C208S}$(SSG) (Fig. 5b, c). The PfGrx$^{C88S}$-catalyzed deglutathionylation of monothiol roGFP2$^{C208S}$(SSG) is therefore two orders of magnitudes faster than the glutathionylation of roGFP2$^{C208S}$(SH) by PfGrx$^{C88S}$(SSG) (Table 1) corresponding to the strongly shifted equilibrium towards glutathionylated PfGrx$^{C88S}$ and reduced roGFP2$^{C208S}$ in the redox mobility shift assays (Fig. S1). The positions of the other equilibria from the redox mobility shift assays in Fig. S1 were also in good agreement with the more balanced ratios of the rate constants for the corresponding (de)glutathionylations from Table 1. As opposed to the reactivities of both glutathionylated PfGrx variants, reduced PfGrx$^{C88S}$ reacted with glutathionylated monothiol roGFP2 about six to eight times faster than PfGrx$^{C32S/C88S}$ (Fig. 5a–c). Thus, the presence of the thiol group of C32 in PfGrx seems to impair the glutathionylation and to facilitate the deglutathionylation of roGFP2. Furthermore, glutathionylated residue C151 of roGFP2 reacted three to four times faster than glutathionylated residue C208, which is again the opposed reactivity as observed for the oxidation kinetics of both monothiol roGFP2 variants (Table 1).

In summary, we established an assay for the direct and continuous monitoring of a glutaredoxin-dependent deglutathionylation reaction using roGFP2(SSG) monothiol variants as model substrates. The residue-specific reactivities and $k_{red}$ values for C151 and C208 in roGFP2(SSG) as well as active-site residue C29 in reduced dithiol PfGrx$^{C88S}$ and monothiol PfGrx$^{C32S/C88S}$ in Fig. 5 correlate inversely with the reactivities and $k_{ox}$ values for the same residues of the reverse reactions in Fig. 3. The thiol group of C32 in PfGrx therefore facilitates the C29-dependent deglutathionylation of roGFP2(SSG) without forming an intramolecular disulfide bond. The $k_{red}$ value of $1.4 \times 10^6\,M^{-1}\,s^{-1}$ for PfGrx$^{C88S}$(SH)$_2$ is about two orders of magnitude higher than the highest $k_{ox}$ value for PfGrx$^{C32S/C88S}$(SSG) in

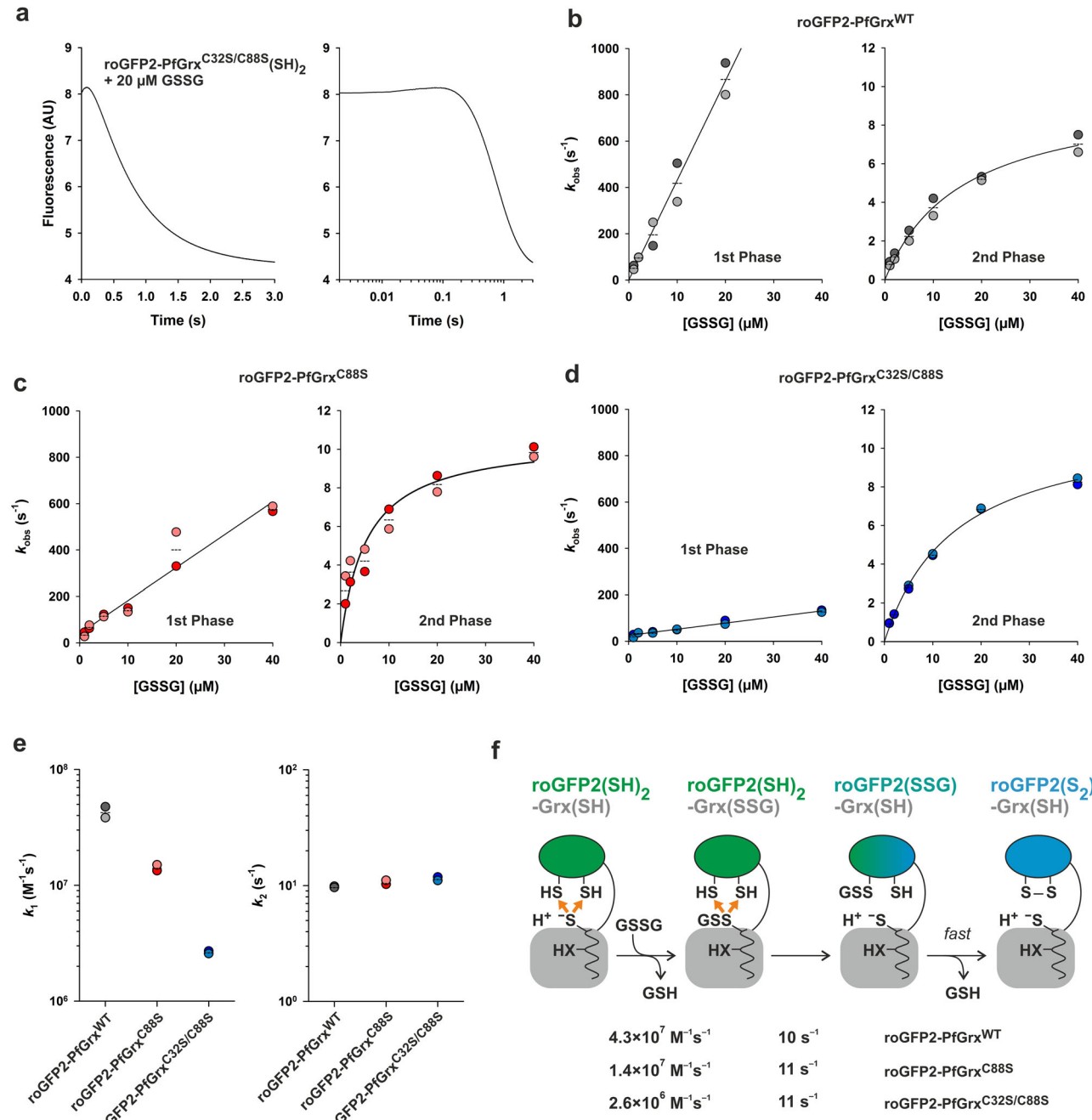

**Fig. 4 | Rate constants for the oxidation of roGFP2$^{WT}$-PfGrx fusion constructs.** **a** Representative biphasic stopped-flow oxidation kinetics at 484 nm for the reaction between 1 μM reduced roGFP2$^{WT}$-PfGrx$^{C32S/C88S}$ and 20 μM GSSG. Secondary plots for the $k_{obs}$ values from the first and second phase of the reaction between GSSG and 1 μM reduced roGFP2$^{WT}$-PfGrx fusion construct for (**b**) PfGrx$^{WT}$, (**c**) PfGrx$^{C88S}$, or (**d**) PfGrx$^{C32S/C88S}$. **e** Overview of the second order rate constants $k_1$ for the first phase (left) and the GSSG-independent maximum rate constants $k_2$ for the second phase (right). The rate constants are also listed in Table 1. **f** Data interpretation of the oxidation kinetics for the three different roGFP2 fusion constructs. The arrows in orange indicate the altered redox environment for roGFP2(SH)$_2$ due to the glutathionylation of the PfGrx moiety. All data sets were generated from at least two independent biological replicates.

accordance with a physiological relevance of glutaredoxins as deglutathionylating enzymes.

**PfGrx-dependent reduction kinetics for roGFP2$^{WT}$(S$_2$)**

To address the previously suggested alternative models for the glutaredoxin-dependent reduction of non-glutathione protein disulfides (Fig. 1c, e)[6,34,60], we conducted a detailed analysis of the reaction between reduced PfGrx variants and roGFP2$^{WT}$(S$_2$) (Figs. 6 and 7, Table 1). The reduction of roGFP2$^{WT}$(S$_2$) required both PfGrx and GSH (Fig. 6) as opposed to the reduction of the glutathionylated monothiol

roGFP2 variants (Fig. 5). The GSH-containing syringe of the stopped-flow spectrofluorometer was supplemented again with 0.4 U/mL GR and 0.25 mM NADPH to remove trace amounts of GSSG and to prevent the thermodynamically favored reverse reaction[32]. The nonenzymatic initial reaction between 1 μM roGFP2$^{WT}$(S$_2$) and physiological millimolar GSH concentrations was too slow to be detectable and to be of any relevance in the presence of glutaredoxins in accordance with previous analyses[32]. Thus, we can exclude a direct uncatalyzed reduction of roGFP2$^{WT}$(S$_2$) by GSH (i.e. mechanism (ii) from Fig. 1e). Neither reduced monothiol PfGrx$^{C32S/C88S}$ nor dithiol PfGrx$^{C88S}$ reacted

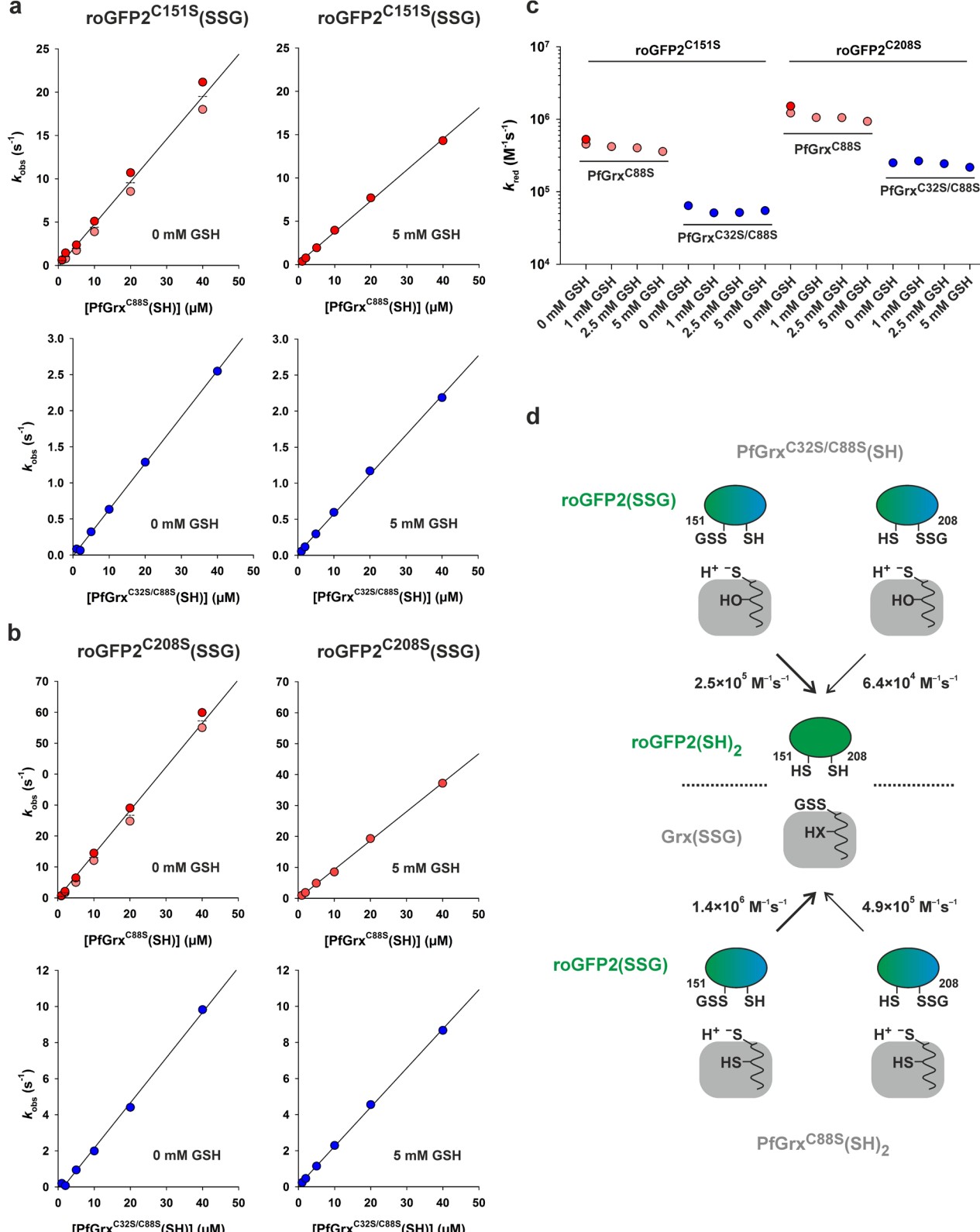

**Fig. 5 | Rate constants for the deglutathionylation of monothiol roGFP2 variants.** Representative secondary plots for the $k_{obs}$ values from the reaction between (**a**) 1 μM roGFP2$^{C151S}$(SSG) or (**b**) 1 μM roGFP2$^{C208S}$(SSG) and reduced PfGrx$^{C88S}$ (top) or monothiol PfGrx$^{C32S/C88S}$ (bottom) in the absence (left) or presence of GSH (right). **c** Overview of the second order rate constants $k_{red}$ for the PfGrx-dependent reduction of the indicated roGFP2(SSG) mutants at four different GSH

concentrations. The rate constants are also listed in Table 1 and Supplementary Table S1. **d** Data interpretation of the reduction kinetics for both monothiol roGFP2(SSG) variants and both PfGrx mutants. The faster reduction pathways are highlighted with thick arrows. All data sets were generated from one or two independent biological replicates.

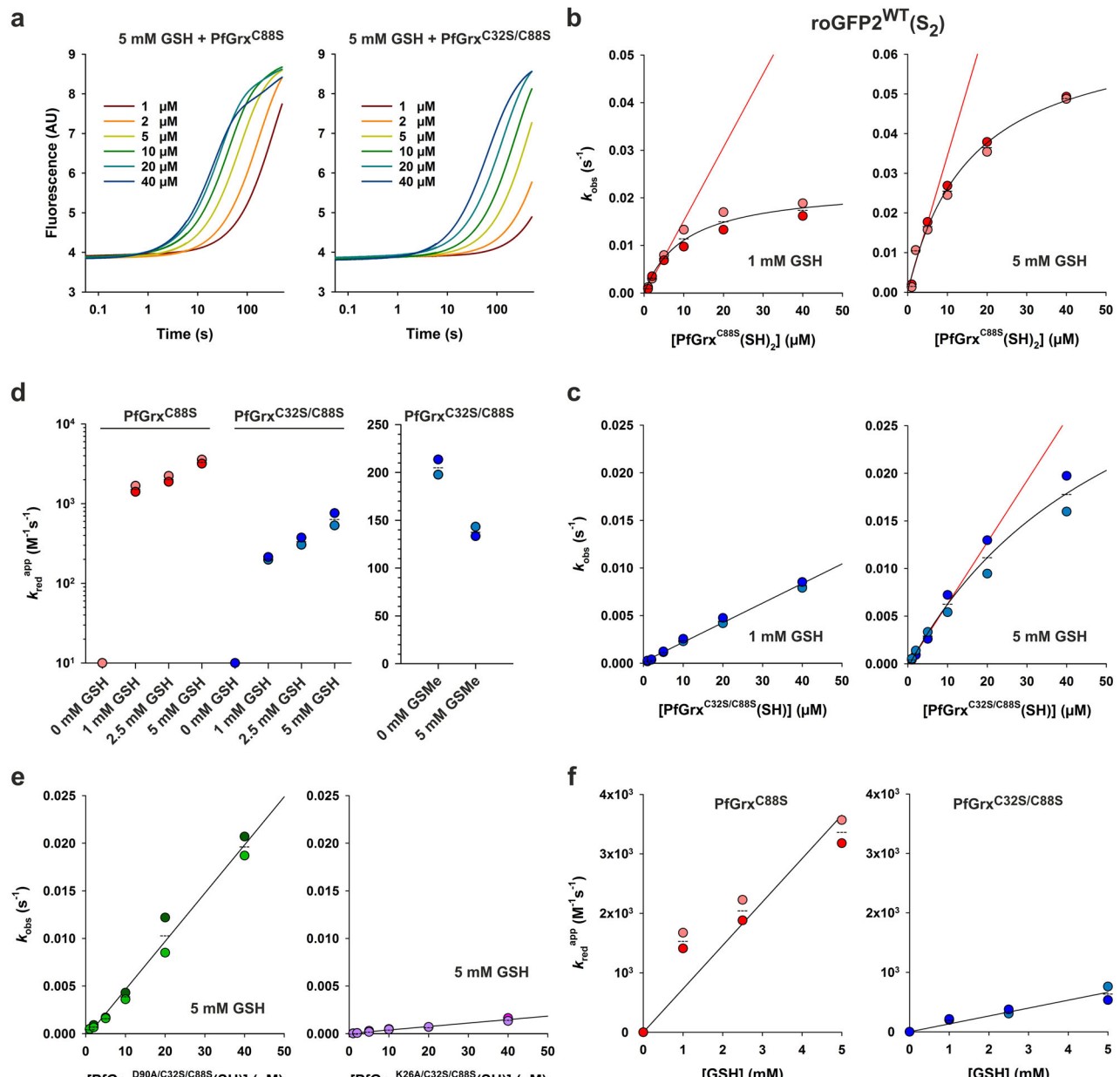

**Fig. 6 | Rate constants for the reduction of roGFP2$^{WT}$(S$_2$). a** Representative biphasic stopped-flow reduction kinetics at 484 nm for the reaction between 1 μM roGFP2$^{WT}$(S$_2$) and reduced dithiol PfGrx$^{C88S}$ (left) or monothiol PfGrx$^{C32S/C88S}$ (right) in the presence of 5 mM GSH. **b, c** Representative secondary plots of the $k_{obs}$ values for the first phase with reduced PfGrx$^{C88S}$ or PfGrx$^{C32S/C88S}$ at 1 mM (left) or 5 mM GSH (right). **d** Overview of the apparent second-order rate constants $k_{red}^{app}$ for the first phase of the PfGrx-dependent reduction of roGFP2(S$_2$) at different GSH concentrations (left) or with 1 mM GSH in the presence or absence of the competitive inhibitor GSMe (right). No activity was detected for 0 mM GSH. The rate constants are also listed in Table 1 and Supplementary Table S1. **e** Secondary plots for the first phase of the reduction kinetics of roGFP2(S$_2$) at 5 mM GSH using either a D90A (left) or K26A (right) mutant of reduced monothiol PfGrx$^{C32S/C88S}$. **f** Tertiary plots of the GSH-dependent $k_{red}^{app}$ values for reduced PfGrx$^{C88S}$ (left) or PfGrx$^{C32S/C88S}$ (right). True $k_{red}$ values were estimated by linear regression and are listed in Table 1. All data sets were generated from at least two independent biological replicates.

with roGFP2$^{WT}$(S$_2$) in the absence of GSH, and the reduction velocities depended on both the PfGrx and GSH concentration (Fig. 6a–d). A rapid direct reaction between reduced PfGrx and roGFP2$^{WT}$(S$_2$) in accordance with the dithiol mechanism from Fig. 1c or the monothiol mechanism (i) from Fig. 1e therefore also seems unlikely. Adequate fits of the biphasic kinetic traces for the reduction of roGFP2$^{WT}$(S$_2$) from Fig. 6a required a double exponential equation. Secondary plots for the rate constants of the rapid first reaction phase at variable PfGrx concentrations deviated from linearity and showed a hyperbolic correlation at millimolar GSH concentrations (Fig. 6b, c). Apparent $k_{red}$ values were estimated from linear fits at low PfGrx concentrations. The

reactions with PfGrx$^{C88S}$ were faster than with PfGrx$^{C32S/C88S}$ and the highest $k_{red}^{app}$ value of $3.4 \times 10^3\,M^{-1}\,s^{-1}$ was detected for PfGrx$^{C88S}$ at 5 mM GSH (Fig. 6d, Table 1). The addition of the competitive inhibitor *S*-methyl glutathione (GSMe) significantly decreased the $k_{red}^{app}$ value (Fig. 6d). GSMe is usually a much weaker inhibitor in the HEDS assay or in assays with GSSR substrates[11,14,17], suggesting that the competition between GSH and GSMe in the roGFP2 measurements occurred at the 1st and not the 2nd glutathione interaction site. No reaction was observed when L-cysteine was used instead of GSH in accordance with a specific substrate recognition (Fig. S3). Furthermore, alanine mutations of the established glutathione-scaffold site residue D90 or of

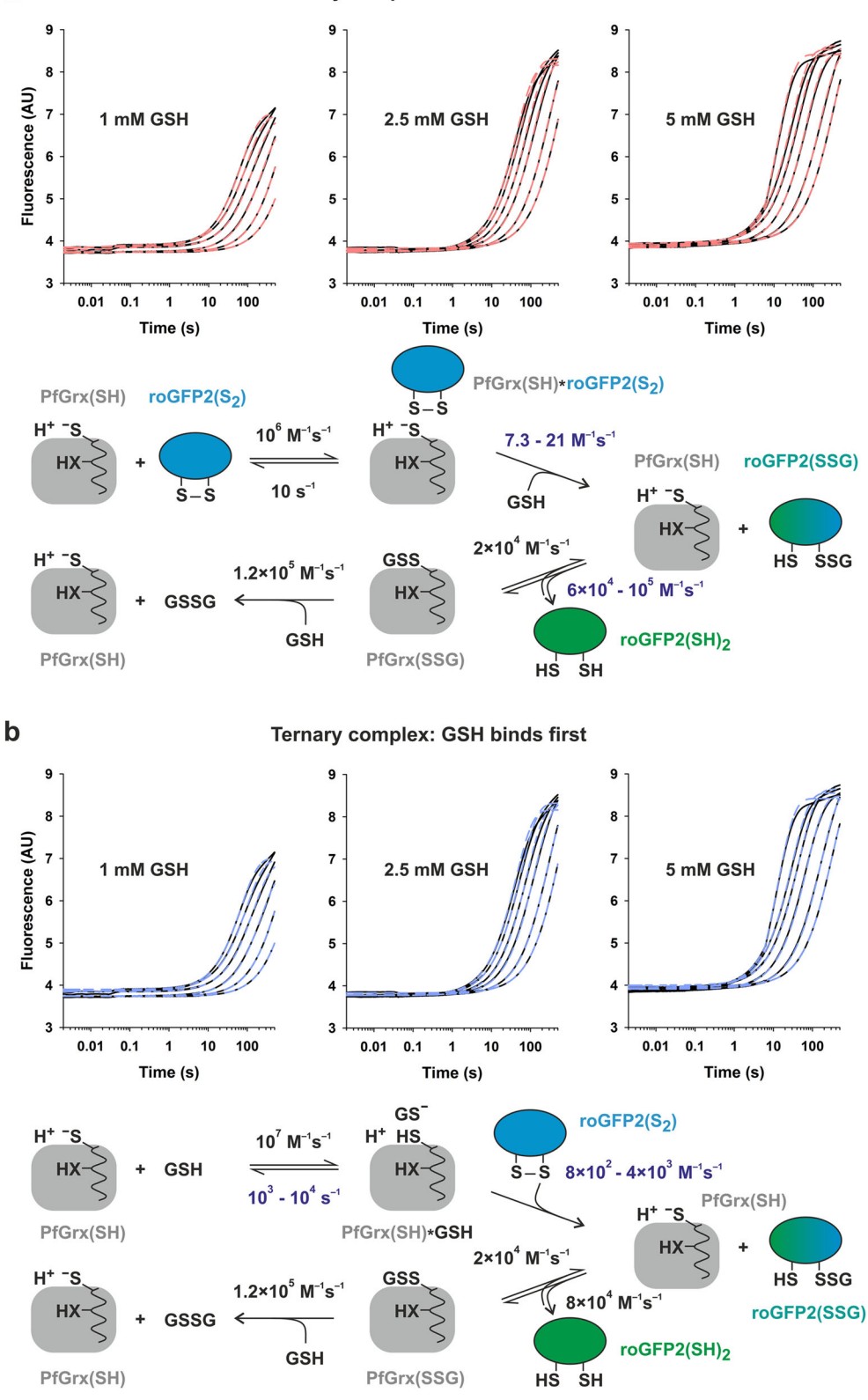

**Fig. 7 | Simulations of the stopped-flow kinetic data for the reduction of roGFP2$^{WT}$(S$_2$).** Simulation of the biphasic kinetic data for the reaction between 1 μM roGFP2$^{WT}$(S$_2$) and 1, 2.5, or 5 mM GSH in the presence of 1–40 μM reduced PfGrx$^{C88S}$. Best fits (dashed lines) were obtained for a reaction within a ternary complex regardless whether (**a**) roGFP2(S$_2$) or (**b**) GSH was first bound to PfGrx. Rounded rate constants from Table 1 in black were chosen as input parameters to calculate the missing parameters in blue. The relative fluorescence intensity of each roGFP2 redox species was taken from Fig. 2.

crucial residue K26 in monothiol PfGrx$^{C32S/C88S}$ slowed down the reaction and resulted in linear secondary diagrams for the first rate constant (Fig. 6e). The $k_{red}^{app}$ values for the D90 and K26 mutants at 5 mM GSH were 78 and 5.6% of the $k_{red}^{app}$ value for PfGrx$^{C32S/C88S}$ as a control. These percentages are almost identical to the percentages for the second-order rate constants of the same mutants with GSSCys as a substrate[14], indicating that GSH also occupies the 1st glutathione-interaction site of PfGrx for the reduction of roGFP2$^{WT}$(S$_2$). Based on the assumption that a ternary complex is formed between PfGrx, GSH, and roGFP2$^{WT}$(S$_2$), we estimated true $k_{red}$ values from tertiary plots at variable GSH concentrations of $7.3 \times 10^5 \, M^{-2} \, s^{-1}$ and $1.3 \times 10^5 \, M^{-2} \, s^{-1}$ for PfGrx$^{C88S}$ and PfGrx$^{C32S/C88S}$, respectively (Fig. 6f).

Finally, we simulated the PfGrx$^{C88S}$-dependent reduction kinetics for roGFP2(S$_2$) using the three alternative mechanisms from Fig. 1e and the rate constants from Table 1 (Fig. 7). We excluded the possibility that the first reaction phase reflects the deglutathionylation of roGFP2$^{WT}$ since this would necessitate an earlier drastic increase of the fluorescence due to the opening of the disulfide bond in roGFP2$^{WT}$(S$_2$). Simulations for alternative monothiol mechanisms (i) and (ii) from Fig. 1e did not reveal appropriate fits and were excluded based on incorrect fluorescence amplitudes and deviating, substrate concentration-dependent rate constants for the data sets at 1, 2.5, and 5 mM GSH. In contrast, good fits with concentration-independent rate constants were obtained for the monothiol mechanism (iii) from Fig. 1e, regardless whether the formation of the ternary complex was initiated by GSH or roGFP2$^{WT}$(S$_2$) (Fig. 7). We therefore suggest a mechanism that can be described according to the Cleland nomenclature as a combination of a random bi-uni mechanism and a ping-pong bi-bi (or uni-uni-uni-uni) mechanism in which the single product of the rate-limiting random mechanism becomes the first substrate of the ping-pong mechanism (Fig. 8a, see also discussion). The small second-order rate constant for the simulated reaction between GSH and the complex of reduced PfGrx$^{C88S}$ and roGFP2$^{WT}$(S$_2$) might reflect a poor accessibility of the 1st glutathione interaction site in this complex (Fig. 7a), whereas the simulated reaction for roGFP2$^{WT}$(S$_2$) was by two orders of magnitude faster when GSH was already bound to PfGrx$^{C88S}$ (Fig. 7b). Whether the roGFP2$^{WT}$(SSG) intermediate is released presumably depends on its size and the geometry of the first and second transition state as outlined below (Fig. 8a, b).

In summary, reduction of roGFP2$^{WT}$(S$_2$) as a model non-glutathione protein disulfide substrate requires the presence of GSH and PfGrx. The kinetic data can be best explained by a ternary complex in which PfGrx recruits and activates GSH at the 1st glutathione-interaction site to facilitate the rate-limiting glutathionylation of roGFP2$^{WT}$(S$_2$) and its subsequent deglutathionylation in accordance with monothiol mechanism (iii) from Fig. 1e (Fig. 8a, b).

## Discussion

The enzymatic formation and reduction of disulfides in physiological systems clearly differs from uncatalyzed redox equilibrations in vitro[5,7,63,78,79]. Since fundamental biochemical pathways evolved in the absence of molecular oxygen, most intracellular disulfides are rather transient and most cysteinyl residues are kept in a reduced state with the help of the thioredoxin or the glutathione/glutaredoxin system[7,63,77]. While in vitro equilibrations between redox couples such as roGFP2 and GSH/GSSG require minutes or even hours to reach a thermodynamic endpoint ($\Delta G = 0$), enzyme-catalyzed thiol-disulfide exchange reactions usually occur very rapidly and either maintain or reach a steady state ($\Delta G \neq 0$)[7,31,32,60,63,67,78,79]. Here we determined the mechanisms and rate constants for the glutathione/glutaredoxin-dependent redox reactions of the widely used redox probe roGFP2 and addressed the question of how mono- and dithiol glutaredoxins catalyze the reduction of roGFP2(S$_2$) as a model non-glutathione protein disulfide substrate. We showed that the PfGrx-dependent oxidation

and reduction of roGFP2 both efficiently occur via a monothiol mechanism (Fig. 8a). The dithiol mechanism was too slow to compete with the monothiol mechanism in the presence of GSH or GSSG and was only detected for the oxidation of roGFP2 using diamide-treated PfGrx without GSH. The latter result confirms the previous study on the ScGrx1-catalyzed oxidation of rxYFP, although the removal of the second active-site cysteinyl residue had opposite effects on the reactivity of PfGrx and ScGrx1 (Table 1)[31]. Furthermore, we established a stopped-flow kinetic assay for the direct continuous detection of the glutaredoxin-dependent deglutathionylation of monothiol roGFP2 variants. The rate constants from our deglutathionylation assays between $6.4 \times 10^4$ and $1.4 \times 10^6 \, M^{-1} \, s^{-1}$ are very similar to the rate constants for PfGrx or plant and mammalian glutaredoxins from discontinuous or coupled enzymatic assays with other high and low molecular weight GSSR substrates (Table 1)[9,11,14,19], thus confirming our assay as an alternative reliable method for the direct analysis of enzyme-catalyzed deglutathionylations. Although residues C32 and C88 of PfGrx are redox-inactive during the monothiol mechanism, they both facilitate the deglutathionylation of GSSR substrates. Analogous residues in other glutaredoxins were also shown to increase[11,20,31,56] or to decrease[18,55–57] their activity. While the second-order rate constants of PfGrx$^{C88S}$ and PfGrx$^{C32S/C88S}$ for GSSG are one order of magnitude higher than for bulky roGFP2(SSG), the rate constant for the glutathionylation of PfGrx$^{C32S/C88S}$ by GSSG is twice that with GSSCys (Table 1)[14]. Thus, not only the size but also the charge and shape of the glutathionylated substrate are relevant with GS$^-$ in GSSG being the optimal leaving group in accordance with a 2nd glutathione-interaction site (Fig. 1b).

What can we learn about the mechanism for the reduction of roGFP2(S$_2$)? Despite the thermodynamically highly favorable reverse reaction, PfGrx$^{C88S}$(S$_2$) reacted very slowly with roGFP2$^{WT}$(SH)$_2$ most likely due to the instability and high energy of the Grx(SSR) intermediate. Thus, it is unlikely that the formation of a Grx(SSR) intermediate is a preferred reaction pathway for the reduction of roGFP2(S$_2$). Our kinetic data indeed support an enzymatic reduction of roGFP2(S$_2$) that requires a ternary complex between the glutaredoxin, GSH, and the non-glutathione disulfide substrate (Fig. 8a, b). The presence of the second active-site cysteinyl residue in PfGrx improves the reactivity of the ternary complex but does not form an intramolecular disulfide bond in contrast to a dithiol mechanism. Glutathionylated roGFP2 is subsequently reduced via the monothiol ping-pong mechanism for GSSR substrates. The enzyme-catalyzed reverse reaction between GSSG or GSSR and reduced roGFP2 releases roGFP2(SSG) which can instantaneously react via a nonenzymatic shortcut yielding GSH and roGFP2(S$_2$) (Fig. 8a). Thus, the enzymatic reduction of roGFP2(S$_2$) by GSH follows monothiol mechanism (iii) and requires a glutaredoxin to form roGFP2(SSG). On the basis of microscopic reversibility of enzymatic reactions, the enzyme-catalyzed oxidation of roGFP2(SH)$_2$ by GSSG or GSSR follows the reverse reaction of mechanism (iii). However, the nonenzymatic formation of roGFP(S$_2$) from roGFP2(SSG) also proceeds extremely rapidly and competes with the final enzyme-catalyzed reaction step with implications for redox measurements as outlined below. The reduction of roGFP2(S$_2$) is therefore a good example for an enzyme using an alternative reaction pathway to overcome a kinetic challenge, and we hypothesize that other non-glutathione disulfide substrates with reactive proximal cysteinyl residues and rather negative redox potentials also require a ternary complex for their reduction. The reaction mechanism for roGFP2(S$_2$) as a model non-glutathione disulfide substrate resembles the first part of the GSH-dependent reduction of the sulfenic acid of human peroxiredoxin 6 that is catalyzed by glutathione transferase P1-1 (which shares several structural features with glutaredoxins)[7,80–82]. The first GSH molecule is activated at the 1st glutathione-interaction site of the glutaredoxin. This step might involve acid-base catalysis by the active-site cysteinyl thiolate (which could explain an effect of the

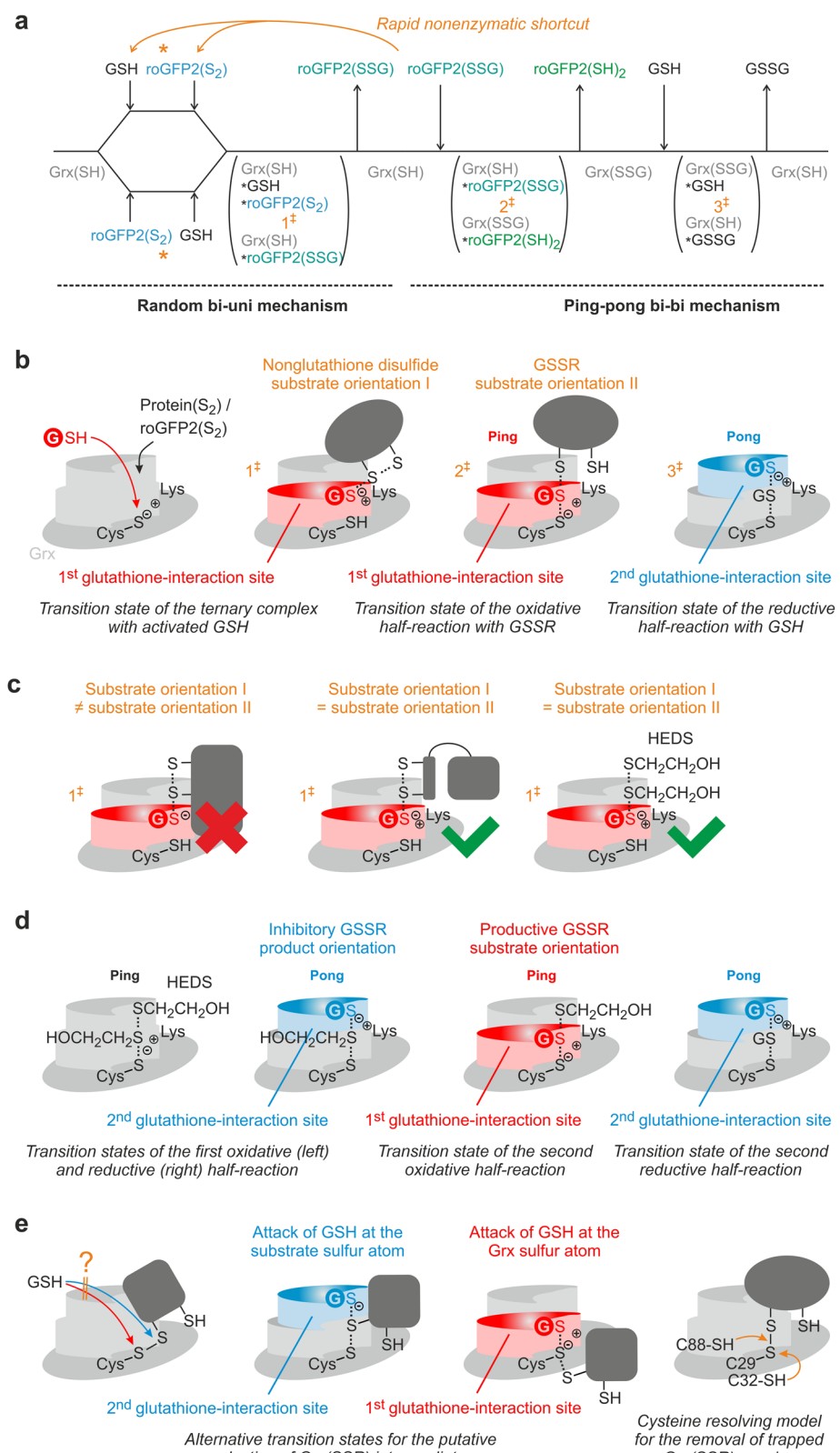

second cysteinyl residue), and/or the conserved lysine residue, yielding GS⁻ in analogy to glutathione transferases (Fig. 8b)[7]. Please note that the previously designated glutathione-activator site for the monothiol mechanism with GSSR substrates corresponds to the 2nd and not the 1st glutathione-interaction site (Fig. 1b)[7,8,13–15]. Since we have now shown that the 1st glutathione-interaction site can serve not only as a scaffold site for GSSR substrates but also as an activator site

for GSH for the reduction of non-glutathione disulfide substrates, we would like to suggest to preferably use the terms 1st and 2nd glutathione-interaction site (e.g., in combination with the function as a scaffold or activator) to avoid ambiguities in the future. The 1st glutathione-interaction site should be frequently (although transiently) occupied by GSH at millimolar concentrations, whereas diverse non-glutathione disulfide substrates are far less abundant

**Fig. 8 | Potential effects of the transition state geometries and substrate sizes on the glutaredoxin-catalyzed reduction of non-glutathione disulfide substrates. a** Cleland scheme of the deduced monothiol reaction mechanism for the PfGrx-catalyzed redox reactions of roGFP2. The reaction with roGFP2(S$_2$) as a model non-glutathione disulfide substrate begins with a rate-limiting random bi-uni mechanism and is shown from left to right. The subsequent deglutathionylation of the roGFP2(SSG) intermediate follows a ping-pong mechanism. The reverse reaction with GSSG (or GSSR) and roGFP2(SH)$_2$ as model protein with surface-exposed cysteinyl residues is shown from right to left and will probably deviate by a nonenzymatic shortcut yielding roGFP2(S$_2$) from roGFP2(SSG). See text for details.

**b** Schematic representations of the three predicted transition states for the reactions from panel a. **c** Potential effects of the size and reaction geometry on the formation and orientation of the first transition state from (**a**). **d** Schematic representations of the four predicted transition states for the suggested reduction of HEDS according to monothiol mechanism (i). **e** Schematic representations for the putative reduction of Grx(SSR) intermediates according to alternative monothiol mechanism (i) are shown on the left and in the middle. If the Grx(SSR) species cannot be attacked by GSH, a second or moderately conserved third cysteinyl residue in glutaredoxins could prevent the accumulation of trapped Grx(SSR) species according to the cysteine resolving model shown on the right.

in vivo and should encounter the active site less frequently. Hence, although future kinetic studies might confirm a random mechanism for the rate-limiting first part of monothiol mechanism (iii) in vitro, GSH is probably the first and the non-glutathione disulfide the second substrate under physiological conditions in vivo (Fig. 8a). Since GSH interacts with the 1st glutathione-interaction site, glutathionylated products of the random bi-uni mechanism might already adopt a suitable orientation as classical GSSR substrates for the subsequent deglutathionylation (Fig. 8b, c) to avoid a futile cycle due to the nonenzymatic shortcut (Fig. 8a). Whether the deglutathionylation requires a release or reorientation of the GSSR intermediate should depend on whether the four sulfur atoms of the disulfide substrate, GSH, and the active-site cysteinyl residue can be arranged like beads on string. While such an arrangement could be achieved by small disulfides such as HEDS, bulky protein substrates might cause significant deviations from this reaction geometry (Fig. 8b, c). We therefore assume that the reaction sequence for the reduction of roGFP2(S$_2$) requires at least a reorientation of the GSSR intermediate and that this may also be the case for other non-glutathione protein disulfide substrates. Whether this hypothesis is correct or whether accessible disulfides (e.g., within an extended loop or a flexible N- or C-terminus) circumvent such steric limitations remains to be analyzed.

What could be the reasons for possible alternative glutaredoxin mechanisms for the reduction of non-glutathione disulfide substrates? While the dithiol mechanism for EcGrx1 and EcRNR or EcPR can be explained by specific protein-protein interactions (Fig. 1c, d)[42,58], a rather unspecific GSH-dependent monothiol mechanism might be better suited for the reduction of a variety of physiological non-glutathione disulfide substrates to maintain metabolites and proteins in a reduced state (Fig. 1e)[34]. Several candidate non-glutathione disulfide substrates for the dithiol mechanism have been identified using monothiol glutaredoxin mutants as bait proteins for the trapping of Grx(SSR) species (Fig. 1c)[83–85]. Non-glutathione disulfide substrates that employ monothiol mechanism (iii) should not be trapped as Grx(SSR) intermediates, which might explain the rather small number of verified non-glutathione disulfide substrates of glutaredoxins to date. A rate-limiting step involving a ternary complex according to Fig. 8a could also explain the previously observed sequential kinetic patterns for the small non-glutathione substrate HEDS (Fig. 8c)[3,7,12,14,15,30]. A non-exclusive alternative explanation for the kinetic patterns with HEDS (and the decreased enzyme activity at high GSH concentrations) could be a Grx(SSCH$_2$CH$_2$OH) intermediate that is attacked by GSH according to monothiol mechanism (i) from Fig. 1e, yielding a GSSR product that has to change its inhibitory orientation before it can be reduced at the 1st glutathione-interaction site (Fig. 8d)[3,14,15,30]. Monothiol mechanism (i) and (iii) remain to be analyzed in more detail for HEDS and small non-glutathione disulfides. Mechanism (ii) from Fig. 1e has also been suggested (and is still commonly propagated) for HEDS as a non-glutathione substrate[9,57]. However, the kinetic patterns do not support a ping-pong mechanism and the nonenzymatic first reaction with a second order rate constant of 0.6 M$^{-1}$ s$^{-1}$ is too slow to explain the rapid turnover[3,12,14,15,30]. Many other nonenzymatic glutathionylations according to mechanism (ii) probably also have such small rate constants[63,86]. The reduction of human RNR, which was suggested to

follow mechanism (ii)[46], therefore needs to be analyzed and monothiol mechanism (iii) should be also considered. A dithiol mechanism according to Fig. 1c and the monothiol mechanism (i) from Fig. 1e were suggested for slow thermodynamic thiol-disulfide equilibrations between different glutaredoxins, GSH, and the metal-binding domains of the proteins HMA4 and Atox1[60]. The suggested dithiol mechanism was based on experiments at a single substrate concentration that required more than 8 h for redox equilibration in the presence of 0.5 µM glutaredoxin with or without 1 µM GSH; conditions that are neither physiological nor exemplary for rapid glutaredoxin catalysis. Furthermore, the relevance of a Grx(SSR) intermediate and an attack by GSH at the glutaredoxin sulfur atom according to monothiol mechanism (i) were postulated for the reduction of the non-glutathione disulfide bond of the HMA4 domain[60]. The slow protein reduction was analyzed at a single substrate concentration in the presence of 0.1 µM glutaredoxin and 0.8 mM GSH. Neither the accessibility of the Grx(SSR) sulfur atoms (Fig. 8e) nor the presence or concentration of the mixed disulfide were addressed. In contrast, a GSSR intermediate was detected and its decreased steady-state concentration at higher glutaredoxin concentrations was suggested to support monothiol mechanism (i)[60]. However, this result is expected considering the very high activity of glutaredoxins as deglutathionylating enzymes and a decreased GSSR concentration neither excludes mechanism (ii) or (iii) from Fig. 1e. We therefore suggest for future studies on monothiol mechanism (i) to take the bulkiness of the putative Grx(SSR) intermediate, an S$_N$2 reaction geometry, and a possible accumulation of Grx(SSR) as an artifact into account (Fig. 8e). For example, if the glutaredoxin sulfur atom in Grx(SSR) species was accessible for a productive reaction with GSH, one would not expect to trap such species by monothiol glutaredoxin mutants[83–85]. We have previously encountered a potentially similar mechanistic misdirection regarding the relevance of a mixed disulfide between PfGrx and the peroxiredoxin PfAOP. While electrophoretic mobility shift assays and mass spectrometry data both suggested a potential role of this mixed disulfide for catalysis[26], kinetic assays later revealed the irrelevance for the catalytic cycle and thus confirmed that the Grx(SSR) species was an artifact[29]. The reduction of roGFP2(S$_2$) according to monothiol mechanism (iii) and previous experiments in yeast also raise the question whether the dithiol mechanism is crucial for the reduction of many protein substrates[34]. At this point it cannot be ruled out that some of the proteins that were captured with monothiol glutaredoxin mutants as Grx(SSR) species might have just accumulated as unspecific dead-end products. Thus, the second and a moderately conserved third cysteinyl residue (e.g. C32 and C88 in PfGrx) could not only form an intramolecular disulfide during catalysis or affect the reactivity of the active-site thiolate but also function as a resolving cysteine for trapped Grx(SSR) species (Fig. 8e)[8,13,26,34]. As outlined above, studies on non-glutathione disulfide substrates unfortunately regularly lack more detailed kinetic analyses, first, to confirm the efficient turnover of the substrate candidates and, second, to validate the dithiol mechanism. For example, control experiments with millimolar GSH and a monothiol glutaredoxin mutant to exclude a monothiol mechanism often appear to be missing[40,41,87]. In summary, monothiol mechanism (iii), as exemplified for roGFP2(S$_2$) in Fig. 8a, b, provides a plausible alternative

mechanism for the reduction of non-glutathione disulfide substrates that could complement the established dithiol mechanism for EcGrx1 and EcRNR. Whether this mechanism is commonly employed for a variety of substrates and whether monothiol mechanisms (i) and (ii) are really relevant for the reduction of other non-glutathione disulfide substrates remains to be analyzed.

What are the implications of our study regarding noninvasive redox measurments? Our roGFP2 measurements confirm (i) that dynamic roGFP2 sensing (or the physiological glutathionylation/deglutathionylation of regular surface-exposed cysteinyl residues) requires a glutaredoxin activity, and (ii) that glutathionylated proteins or metabolites should be very transient redox species when GSH and glutaredoxins are present (unless they are stabilized by a kinetic uncoupling mechanism)[5,32,34,35,61-64]. The very low rate constants for direct, glutathione-independent dithiol-disulfide exchange reactions between glutaredoxins and roGFP2 or other protein($S_2$)/protein$(SH)_2$ couples (despite favorable thermodynamics) are a prerequisite and great advantage for reliable redox measurements in vivo. If Grx(SSR) intermediates and mixed disulfides with roGFP2 were more stable and were readily formed, glutaredoxins would sense and transfer a plethora of other redox states to roGFP2 probes. Regarding intracellular roGFP2-based structure-function analysis of glutaredoxins, we now showed that the GSSG-dependent oxidation of the fused glutaredoxin moiety (and not the glutathionylation of the roGFP2 moiety) is rate-limiting unless saturating GSSG concentrations are present. Such high micromolar GSSG concentrations are unlikely to occur in most sub-cellular compartments but could be reached in the secretory pathway or during titration experiments of transgenic cell lines with an upregulated GSSG transporter[35,63,64,77]. This explains why intracellular oxidation kinetics for roGFP2 fusion constructs with glutaredoxin mutants showed excellent correlation with the second order rate constants of the oxidative half-reaction from in vitro experiments[15,26,34,35]. No saturation behavior for GSSG was previously detected for the fusion construct between rxYFP and ScGrx1[31], which makes sense considering that PfGrx and many other glutaredoxins are at least two orders of magnitude more active than ScGrx1 (Table 1)[14,56]. One anomaly from previous measurements was the high intracellular steady-state oxidation of roGFP2 for fusion contructs with ScGrx7, in particular, when a WP-motif or a loop from class II glutaredoxins was introduced. In principle, all reactions in Fig. 8a are reversible. However, the negative redox potential of roGFP2, the instantaneous non-enzymatic formation of roGFP2($S_2$) from roGFP2(SSG), and the very slow nonenzymatic reaction of roGFP2($S_2$) with GSH make the non-enzymatic shortcut from Fig. 8a practically irreversible and independent of the fused glutaredoxin. Furthermore, the reduction of roGFP2($S_2$) is the slowest of the enzymatic reaction steps in Fig. 8a (Table 1). We therefore suggest that the intracellular steady-state oxidation of roGFP2 predominantly reflects the ability of the fused glutaredoxin to form a productive ternary complex with GSH and roGFP2($S_2$). This hypothesis can now be tested in a follow-up study.

## Methods
### Materials
Diamide, dithiothreitol (DTT), ethylenediaminetetraacetic acid (EDTA), GSH, GSSG, and yeast GR were from Sigma-Aldrich, L-cysteine was from Carl Roth, isopropyl-β-D-1-thiogalactopyranoside was from Serva, methyl ether poly(ethylene glycol)$_{24}$ maleimide (mmPEG) was from Iris Biotech, and NADPH was from Gerbu. PCR primers were purchased from Metabion. All restriction enzymes, Phusion DNA polymerase, and T4 DNA ligase were from New England Biolabs (NEB).

### Cloning and site-directed mutagenesis
All plasmids and primers for cloning and site-directed mutagenesis are listed as separate Supplementary Data. Fusion constructs *ROGFP2^WT^-PFGRX^WT^*, *ROGFP2^WT^-PFGRX^C88S^*, and *ROGFP2^WT^-PFGRX^C32S^* were PCR-amplified with Phusion DNA polymerase using the corresponding p416TEF plasmids as a template[35] and subcloned into the *Kpn*I and *Avr*II restriction sites of pET45b in strain XL1-Blue. Point mutations in *ROGFP2^WT^-PFGRX^C32S/C88S^*, *ROGFP2^C151S^* or *ROGFP2^C208S^* were introduced by PCR with *Pfu* polymerase using *ROGFP2^WT^-PFGRX^C32S^*/pET45b or *ROGFP2*/pET28a as a template. The template DNA was then digested with *Dpn*I and the plasmids were cloned in strain XL1-Blue. Correct mutations and sequences were confirmed for all constructs by sequencing both strands.

### Heterologous expression and sample preparation
Recombinant C-terminally LEH$_6$-tagged wild-type and mutant roGFP2, N-terminally MRGSH$_6$GS-tagged PfGrx variants, and N-terminally MAH$_6$VGT-tagged roGFP2^WT^-PfGrx^WT^, roGFP2^WT^-PfGrx^C88S^, and roGFP2^WT^-PfGrx^C32S/C88S^ were produced in *E. coli* strain BL21 (DE3) or strain XL1-Blue and purified by Ni-NTA affinity chromatography as described previously[14,26,35,88]. Similar yields and purities were obtained when the recombinant proteins were produced for 4 h at 30 °C in *E. coli* strain SHuffle T7 express (NEB) following induction with 0.5 mM isopropyl-β-D-1-thiogalactopyranoside at an OD of 0.5. Cultures were subsequently cooled in an ice water bath for 15 min and centrifuged (4.000*g*, 15 min, 4 °C). Cell pellets were resuspended in ice-cold buffer 1 containing 20 mM imidazole, 50 mM Na$_x$H$_y$PO$_4$, 300 mM NaCl, pH 8.0 at 4 °C and stirred on ice with DNase I and 10 mg/mL lysozyme for 1 h before sonication. The lysates were centrifuged (10.000*g*, 30 min, 4 °C) and the supernatants were loaded on equilibrated Ni-NTA agarose columns (Qiagen). The columns were washed with 15 column volumes of buffer 1 followed by the elution with 200 mM imidazole, 50 mM Na$_x$H$_y$PO$_4$, 300 mM NaCl, pH 8.0 at 4 °C. The purity of all proteins was confirmed by analytical SDS-PAGE and the concentrations were determined spectrophotometrically at 280 nm.

### Stopped-flow kinetic measurements
For all stopped-flow measurements, freshly purified proteins were reduced with 5 mM DTT on ice for 30 min. Excess DTT and imidazole were removed on a PD-10 desalting column (Merck), and the reduced proteins were eluted with 3.5 mL ice-cold assay buffer containing 100 mM Tris/HCl, 1 mM EDTA, pH 8.0. Protein concentrations were again determined spectrophotometrically at 280 nm. Glutathionylated roGFP2^C151S^, roGFP2^C208S^, PfGrx^C88S^, and PfGrx^C32S/C88S^ were produced by incubation of the reduced enzymes with 5 mM GSSG for 60 min on ice. Oxidized roGFP2^WT^, PfGrx^C88S^, and PfGrx^C32S/C88S^ were produced by incubation of the reduced enzymes with 5 mM Diamide for 60 min on ice. The glutathionylated/oxidized enzymes were again purified via Ni-NTA affinitiy chromatograpghy followed by desalting on a PD-10 column as described above. Stopped-flow measurements were performed at 25 °C in a thermostatted SX-20 spectrofluorometer (Applied Photophysics). The change of fluorescence was measured for up to 600 seconds after mixing (total emission at an excitation wavelength of 400 or 484 nm with a slit width of 2 mm). The oxidation of roGFP2 was investigated by mixing 2 μM of reduced roGFP2 in syringe 1 with variable concentrations of glutathionylated or diamide-oxidized PfGrx in syringe 2. For the reduction of roGFP2, 2 μM diamide oxidized roGFP2^WT^ or 2 μM glutathionylated roGFP2^C151S^ or roGFP2^C208S^ in syringe 1 were mixed with variable concentrations of reduced PfGrx and GSH with 0.4 U/mL GR and 0.25 mM NADPH in syringe 2. The oxidation of the fusion-constructs was followed by mixing 2 μM reduced protein in syringe 1 with variable concentrations of GSSG in syringe 2. The traces of three consecutive measurements were averaged and fitted by single exponential or double exponential regression using the Pro-Data SX software (Applied Photophysics). Rate constants $k_{obs}$ were plotted against the substrate concentration in SigmaPlot 13.0 to obtain second order rate constants from the slopes of the linear fits or first-order rate constants from hyperbolic fits. The Pro-KIV 1.01 software (Applied

Photophysics) was used to perform kinetic simulations of the mechanistic models using the determined rate constants from Table 1 based on the indicated reaction schemes.

## Redox mobility shift assays

For the redox mobility shift assays, proteins were purified, reduced and oxidized as described above. The glutathionylation or oxidation of roGFP2 was investigated by mixing reduced roGFP2$^{C151S}$ or roGFP2$^{C208S}$ with mmPEG in the presence of 5 mM GSSG or Diamide. To monitor the oxidation of roGFP2$^{WT}$ by the intramolecular disulfide of PfGrx$^{C88S}$, 1 µM reduced roGFP2$^{WT}$ was mixed with 2 µM diamide-oxidized PfGrx$^{C88S}$ at room temperature. After different time points 160 µM mmPEG was added and the solution was incubated on ice for 15 min. Proteins were precipitated by adding four volumes of ice-cold acetone and subsequent centrifugation (21.000$g$, 20 min, 4 °C). The supernatant was discarded and samples were resuspended in 5 × Laemmli buffer. To monitor the transfer of the glutathione moiety from PfGrx to roGFP2, 5 µM reduced roGFP2$^{WT}$, roGFP2$^{C151S}$ or roGFP2$^{C208S}$ were mixed with variable concentrations of glutathionylated PfGrx$^{C88S}$ or PfGrx$^{C32S/C88S}$ at room temperature for 1 h. The reaction was quenched by the addition of 5 × Laemmli and 400 µM mmPEG and incubation for 15 min on ice. Samples in Laemmli buffer were boiled at 95 °C for five min and applied to a 15% SDS gel, separated by SDS-PAGE, and stained with coomassie. Uncropped and unprocessed scans of Fig. S1 are provided in the Source Data file.

## Reporting summary

Further information on research design is available in the Nature Portfolio Reporting Summary linked to this article.

## Data availability

The authors declare that all data supporting the findings of this study are provided in the Manuscript and its Supplementary Information/ Source Data files. The data regarding electrophoretic mobility shift assays, nonenzymatic reaction between roGFP2$^{WT}$(SH)$_2$ and GSSG, thiol-specificity of the PfGrx-catalyzed reduction of roGFP2$^{WT}$(S$_2$), and rate constants at variable GSH concentrations are provided in the Supplementary Information. Uncropped and unprocessed scans of Supplementary Fig. 1 are provided in the Source Data file. A list of primers and plasmids are provided as separate Supplementary Data 1. The structures referenced in Fig. 1d are available at the following PDB accession codes: 1EGO, 1GRX, 1QFN and 1EGR. Source data are provided with this paper.

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

## Acknowledgements

This work was funded by the DFG grant DE 1431/20-1 to M.D. and, in part, by DFG RTG 2737 (STRESSistance). We thank Marisa Jakobs for the help with initial redox mobility shift assays and Robin Schumann for cloning wild-type and mutant *ROGFP2*.

## Author contributions

F.G. and L.L. performed the stopped-flow kinetic experiments and analyzed the data. B.H. performed the redox mobility shift assays. M.D. and B.M. conceptualized the study. M.D. supervised the study and wrote the manuscript. All authors read and approved the final version of the manuscript.

## Funding

## Competing interests

The authors declare no competing interests.
