## [Peer Review File · Nature Communications]

REVIEWERS' COMMENTS

Reviewer #1 (Remarks to the Author):

To the Authors

Overall: This is a well-conceived, well-executed, and clearly presented study that definitively resolves a long-term ambiguity in understanding the mechanism of catalysis of reduction of non-glutathionyl protein disulfides by class 1 glutaredoxins. A novel stopped-flow assay developed by the authors was applied to determining the second-order rate constants for the forward and reverse redox reactions of several redox states of the enzyme and substrate components (separate or linked) of the fused enzyme-substrate complex (PfGrx-roGFP2) in the absence and presence of GSH and/or GSSG. Not only are the results and rigorous nature of this study interesting and instructive to enzymologists in general and to investigators of thiol-disulfide redox reactions and enzymes, but also to the broad audience interested in the regulation of intracellular redox homeostasis in health and disease, as the pfGrx-roGFP2 construct is an important tool for probing changes in the intracellular redox environment; and the data presented add to the documentation of the utility of this construct. The manuscript is developed in a systematic fashion with a wealth of background information presented clearly and documented effectively by appropriate citations. The data are presented and discussed in a clear fashion, including appropriate caveats concerning any shortcomings. The conclusions are soundly supported by the data. However there is critical issue that arises in comparison and discussion of the data for catalysis of the forward and reverse reactions, which appears to deviate from the fundamental principle of microscopic reversibility in enzyme catalysis. Specific comments are provided below.

Specific comments:

There are several key findings and insights provided by this study, as noted below.

1. This study provides a rigorous distinction between the monothiol vs. dithiol mechanisms of PfGrx-catalyzed oxidation and reduction of the widely used redox probe roGFP2, serving as a model non-glutathione protein disulfide substrate.
2. This report establishes a novel stopped-flow assay for Grx catalysis. In this regard it is noteworthy that the rate constants determined with this set up are very similar to the rate constants for PfGrx or plant or mammalian Grxs obtained with discontinuous or coupled assays with other GSSR substrates (Table 1). It is remarkable that the rate constants for cys-SSG (the simplest model of protein-SSG) were found to be essentially the same as those for roGFP2-SSG, consistent with the common rate-limiting step being release of Grx-S- as the leaving group.
3. This manuscript provides an excellent critical review in the Introduction and Discussion sections which conveys key insights about the state of understanding of the mechanisms of catalysis by the class I glutaredoxins, acting on glutathionyl and non-glutathionyl disulfides.

Comment: It would be useful for the authors to convey the rationale for choosing PfGrx as the model/representative for the class I glutaredoxins. How many of the key mechanistic findings with PfGrx have been replicated with other Grxs, including mammalian?

Comment: Consider the following statements that might be misinterpreted as being contrary to the concept of microscopic reversibility for enzyme catalysis.

Results, p.13, paragraph 3, “the PfGrx-catalyzed deglutathionylation of monothiol roGFP2 is therefore up to two orders of magnitude faster than the glutathionylation (Table 1).”

Discussion, p. 18, “Thus, the enzymatic reduction of roGFP2(S₂) by GSH follows monothiol mechanism (iii) whereas enzyme-catalyzed oxidation of roGFP2(SH)₂ by GSSG or GSSR follows the reverse reaction of mechanism (ii) ...” This statement seems to suggest that the forward and reverse reactions catalyzed by the same enzyme occur *via* different mechanisms.

In a previous article cited in this manuscript (ref. 5, Gallogly *et al.* 2009), the approach to equilibrium in the presence and absence of hGrx1 from both directions [(GSH + HEDS +/- Grx) vs. (GSSG + βME +/- Grx)] was documented for the interconversion of the non-glutathionyl substrate HEDS and its reduced form βME (see ref. 5, Fig. 2). In that case, the same position of equilibrium was reached when starting from either direction; and the rate enhancement by hGrx1 was the same in both directions. *Has the approach to equilibrium of the redox pair roGFP2(SH₂)/roGFP2(S₂) with GSH or GSSG been studied in the absence and presence of PfGrx; *i.e.*, analogous to the HEDS/βME example described above?

The authors are requested to clarify the meaning of the two statements quoted above and reconcile them with the fundamental principles of catalysis and equilibrium, as necessary.

Minor considerations:

There are a few typos.

Results, p.8, 2nd-last line: “fluorescence”

Materials and Methods, p. 26, paragraph 2: “acetone”

Reviewer #2 (Remarks to the Author):

The study “Deciphering the mechanism of glutaredoxin-catalyzed roGFP2 redox sensing reveals a ternary complex with glutathione for protein disulfide reduction” by Geissel et al. provides a detailed insight into the molecular mechanism and kinetics of the Grx-catalyzed oxidation and reduction of roGFP2. Although roGFP fusion constructs have a broad application in detecting intracellular redox potentials, the underlying enzyme mechanism that facilitate its reduction and oxidation was still not fully unraveled. Using glutaredoxin (Grx) from *Plasmodium falciparum* as model enzyme, the work provides a very detailed mechanistic insight into the oxidation and reduction of roGFP. Vice versa, oxidized roGFP(S₂) served as an excellent tool to analyze the mechanism of the glutathione/glutaredoxin system in reducing non-glutathione disulfides. In the first part of the manuscript, the authors determined oxidation kinetics for reduced roGFP2WT (wild-type) in comparison to the single cysteine mutants, roGFP2C125S and roGFP2C208S and clearly showed that PfGrx efficiently catalyzes the oxidation of roGFP via a glutathione-dependent monothiol mechanism. The fusion construct between roGFP2WT and PfGrx variants further accelerates roGFP oxidation. Although this connection has been reported previously, the authors provide here the underlying kinetics of the reaction mechanism under physiological glutathione/glutathione disulfide concentration. The second part focuses on the PfGrx-catalyzed reduction of roGFP2. The authors determined the reduction kinetics for PfGrx-mediated deglutathionylation roGFP via a monothiol mechanism and further analyzed the reduction kinetics of roGFPWT(S₂) as a non-glutathione disulfide substrate. Conducting detailed analyses of the reactions between reduced PfGrx variants and roGFPWT(S₂), they revealed that the reduction follows a monothiol mechanism under formation of a ternary complex with PfGrx and GSH.

The article is well-written and the observed kinetic data is well-explained in the text. I particularly appreciated the schematic representations included into individual figures to illustrate the mechanistic models, which helped to understand their conclusion. Overall, the work represents a carefully conducted and very detailed kinetic analyses. The molecular mechanisms allow to better understand and evaluate intracellular studies based on roGFP as a redox sensor and provided novel mechanistic insight into the role of Grx in catalyzing the reduction of non-glutathione disulfides in the cell.

I have few minor comments that should be addressed by the authors:

In the abstract, the authors highlight that they initially analyzed PfGrx-catalyzed oxidation of roGFP2. Consistently, the “Results” part starts with the respective experiments. However, in the introduction, the authors did not specifically address the role of Grx in catalyzing the oxidation of protein substrates, such as roGFP2, but mainly focus on Grx-catalyzed reduction.

Regarding Figure 2a, it was not clear to me, why roGFP2WT (SH)₂ was incubated with PfGrxC32S/C88S(SSG) to monitor oxidation, but both roGFP2 mutant variant, roGFP2C215S(SH) and roGFP2C208S(SH), were both treated with GSSG in the absence of Grx.

p 8, l. 11: The authors further claim that the slight changes detected for glutathionylated roGFP2C215S and roGFP2C208S are significant, but they do not provide any statistical test to confirm their statement.

In this regard, I do see the small changes in the spectrum of roGFP2C208 upon GSSG treatment as reflected by an increased fluorescence at an excitation of 400 nm and a decrease at 484 nm (Figure 2a, right). However, I do not see the same behavior for roGFP2C151S. If there is any significant change, the fluorescence decreased at both excitation wavelength upon GSSG treatment (Figure 2a, middle). Also, the stopped-flow measurements showed the decrease at both wavelength upon glutathionylation of roGFP2C151S (Figure 2b, middle) in contrast to roGFP2WT and roGFP2C208S. In the text, the authors did not address this observation, but instead write that “the spectral changes at both excitation wavelengths followed identical kinetics [...] for the formation of roGFP2WT(S2), roGFP2C151S(SSG), or roGFP2C208S(SSG) using PfGrxC32S(C88S(SSG) as an oxidant” (p8, l. 17-19). From the data shown in Figure 2a and b for roGFP2C151S, I do not understand how this conclusion has been drawn and would require some more explanation.

Point-by-point response to the reviewers' comments

Reviewer #1 (Remarks to the Author):

Overall: This is a well-conceived, well-executed, and clearly presented study that definitively resolves a long-term ambiguity in understanding the mechanism of catalysis of reduction of non-glutathionyl protein disulfides by class 1 glutaredoxins. A novel stopped-flow assay developed by the authors was applied to determining the second-order rate constants for the forward and reverse redox reactions of several redox states of the enzyme and substrate components (separate or linked) of the fused enzymesubstrate complex (PfGrx-roGFP2) in the absence and presence of GSH and/or GSSG. Not only are the results and rigorous nature of this study interesting and instructive to enzymologists in general and to investigators of thiol-disulfide redox reactions and enzymes, but also to the broad audience interested in the regulation of intracellular redox homeostasis in health and disease, as the pfGrx-roGFP2 construct is an important tool for probing changes in the intracellular redox environment; and the data presented add to the documentation of the utility of this construct. The manuscript is developed in a systematic fashion with a wealth of background information presented clearly and documented effectively by appropriate citations. The data are presented and discussed in a clear fashion, including appropriate caveats concerning any shortcomings. The conclusions are soundly supported by the data. However there is critical issue that arises in comparison and discussion of the data for catalysis of the forward and reverse reactions, which appears to deviate from the fundamental principle of microscopic reversibility in enzyme catalysis. Specific comments are provided below.

Reply: We thank John J. Mieyal for his careful review, for his kind words, and for pointing out the critical issue of microscopic reversibility, which we have now addressed in more detail in our revised manuscript.

Specific comments:

There are several key findings and insights provided by this study, as noted below.

1. This study provides a rigorous distinction between the monothiol vs. dithiol mechanisms of PfGRx-catalyzed oxidation and reduction of the widely used redox probe roGFP2, serving as a model non-glutathione protein disulfide substrate.
2. This report establishes a novel stopped-flow assay for Grx catalysis. In this regard it is noteworthy that the rate constants determined with this set up are very similar to the rate constants for PfGrx or plant or mammalian Grxs obtained with discontinuous or coupled assays with other GSSR substrates (Table 1). It is remarkable that the rate constants for cys-SSG (the simplest model of protein-SSG) were found to be essentially the same as those for roGFP2-SSG, consistent with the common rate-limiting step being release of Grx-S- as the leaving group.
3. This manuscript provides an excellent critical review in the Introduction and Discussion sections which conveys key insights about the state of understanding of the mechanisms of catalysis by the class I glutaredoxins, acting on glutathionyl and non-glutathionyl disulfides.

Reply: Thank you very much.

Comment: It would be useful for the authors to convey the rationale for choosing PfGrx as the model/representative for the class I glutaredoxins. How many of the key mechanistic findings with PfGrx have been replicated with other Grxs, including mammalian?

Reply: We added the following sentence to clarify why we chose PfGrx as a model enzyme: "PfGrx shares 39% sequence identity with human Grx1, including a CPYC motif and a semiconserved GGC motif, and is well characterized *in vitro* and in roGFP2 assays in yeast.^{14,26,30,35}"

The de/glutathionylation reactions and the monothiol mechanism have been characterized for several Grx isoforms, including mammalian Grx, before. For example, as stated in the manuscript, "The rate constants from our deglutathionylation assays between 6.4×10^4 and $1.4 \times 10^6 \text{ M}^{-1}\text{s}^{-1}$ are very similar to the rate constants for PfGrx or plant and mammalian glutaredoxins from discontinuous or coupled enzymatic assays with other high and low molecular weight GSSR substrates (Table 1).^{9,11,14,19}"

We would expect that the major mechanistic insight from our study regarding protein disulfide reduction can also be transferred to other dithiol class I glutaredoxins. However, experience has taught us that different Grx isoforms are good for surprises, which is why we have refrained from too bold generalizations and preferred more cautious wordings, such as "In summary, monothiol mechanism (iii), as exemplified for roGFP2(S₂) in Fig. 8a,b, provides a plausible novel mechanism for the reduction of non-glutathione disulfide substrates that could complement the established dithiol mechanism for EcGrx1 and EcRNR. Whether this mechanism is commonly employed for a variety of substrates and whether monothiol mechanisms (i) and (ii) are really relevant for the reduction of other non-glutathione disulfide substrates remains to be analyzed." Our future plan is to scrutinize the proposed mechanism with roGFP2 from Fig. 8a with alternative Grx isoforms (including mammalian enzymes) and mutants in a follow-up study.

Comment: Consider the following statements that might be misinterpreted as being contrary to the concept of microscopic reversibility for enzyme catalysis.

Results, p.13, paragraph 3, "the PfGrx-catalyzed deglutathionylation of monothiol roGFP2 is therefore up to two orders of magnitude faster than the glutathionylation (Table 1)."

Reply: We fully agree that the Grx-catalyzed reactions should be reversible and would like to clarify this important aspect to avoid any misinterpretation. According to a reversible half-reaction, the ratio between the rate constants for the forward (glutathionylation) and reverse (deglutathionylation) reaction should reflect the equilibrium constant $K = k_f/k_r$ of the redox couple. To check the validity of our statement, we now compared the estimated ratio between the reduced and glutathionylated monothiol roGFP2 variant at equilibrium (following incubation with our (de)glutathionylated PfGrx variants) from Fig. S1 and the rate constants for the according forward and reverse reaction for the reversible glutathionylation of roGFP2 from Table 1. Despite the obvious limitations of the semiquantitative mobility shift assays, we found a good correlation between the data from Fig. S1 and the ratio of the rate constants:

Involved species	De glutathionylation k_r	Glutathionylation k_f	Ratio $k_r:k_f$	Ratio in gel at equilibrium
PfGrx ⁸⁸ + roGFP ^{C151S}	4.9x10 ⁵	2.1x10 ⁴	23:1	ca. 5:1
PfGrx ^{DM} + roGFP ^{C151S}	6.4x10 ⁴	3.5x10 ⁴	2:1	ca. 1:1
PfGrx ⁸⁸ + roGFP ^{C208S}	1.4x10 ⁶	9.9x10 ³	140:1	>> 20:1
PfGrx ^{DM} + roGFP ^{C208S}	2.5x10 ⁵	1.8x10 ⁴	14:1	ca. 12:1

For the sake of clarity, we have now changed the quoted sentence as follows: "The PfGrx^{C88S}-catalyzed de glutathionylation of monothiol roGFP2^{C208S}(SSG) is therefore two orders of magnitudes faster than the glutathionylation of roGFP2^{C208S}(SH) by PfGrx^{C88S}(SSG) (Table 1) corresponding to the strongly shifted equilibrium towards glutathionylated PfGrx^{C88S} and reduced roGFP2^{C208S} in the redox mobility shift assays (Fig. S1). The positions of the other equilibria from the redox-mobility shift assays in Fig. S1 were also in good agreement with the more balanced ratios of the rate constants for the corresponding (de)glutathionylations from Table 1."

Discussion, p. 18, "Thus, the enzymatic reduction of roGFP2(S₂) by GSH follows monothiol mechanism (iii) whereas enzyme-catalyzed oxidation of roGFP2(SH)₂ by GSSG or GSSR follows the reverse reaction of mechanism (ii)" This statement seems to suggest that the forward and reverse reactions catalyzed by the same enzyme occur *via* different mechanisms.

Reply: The enzyme-catalyzed forward and reverse reactions should indeed follow exactly the same mechanism. However, while the GSH-dependent reduction of roGFP(S₂) to roGFP(SSG) requires the presence of a glutaredoxin to occur efficiently, the formation of roGFP(S₂) from roGFP2(SSG) can also occur efficiently nonenzymatically via a competing uncatalyzed reaction step (Fig. 8a). We modified the quoted sentence accordingly to make this more clear: "Thus, the enzymatic reduction of roGFP2(S₂) by GSH follows monothiol mechanism (iii) and requires a glutaredoxin to form roGFP2(SSG). On the basis of microscopic reversibility of enzymatic reactions, the enzyme-catalyzed oxidation of roGFP2(SH)₂ by GSSG or GSSR follows the reverse reaction of mechanism (iii). However, the nonenzymatic formation of roGFP(S₂) from roGFP2(SSG) also proceeds extremely rapidly and competes with the final enzyme-catalyzed reaction step with implications for redox measurements as outlined below."

In a previous article cited in this manuscript (ref. 5, Gallogly *et al.* 2009), the approach to equilibrium in the presence and absence of hGrx1 from both directions [(GSH + HEDS +/- Grx) vs. (GSSG + βME +/- Grx)] was documented for the interconversion of the non-glutathionyl substrate HEDS and its reduced form βME (see ref. 5, Fig. 2). In that case, the same position of equilibrium was reached when starting from either direction; and the rate enhancement by hGrx1 was the same in both directions. *Has the approach to equilibrium of the redox pair roGFP2(SH₂)/roGFP2(S₂) with GSH or GSSG been studied in the absence and presence of PfGrx; *i.e.*, analogous to the HEDS/βME example described above?

Reply: We could not perform an analogous equilibration experiment for roGFP2 because the equilibrium for the roGFP(SH)₂/roGFP(S₂) couple is far on the side of the disulfide species. As reported previously (Meyer et al. 2007 Plant J), even trace amounts of GSSG in GSH samples result in the formation of roGFP(S₂). We therefore had to perform all experiments with GSH in the presence of glutathione reductase and NADPH to constantly remove GSSG. Our future plan is to study the analogous (PfGrx-catalyzed) equilibrations with GSH and GSSG using a roGFP variant with a much more positive redox potential to avoid this drawback in a follow-up study.

The authors are requested to clarify the meaning of the two statements quoted above and reconcile them with the fundamental principles of catalysis and equilibrium, as necessary.

Reply: We now addressed and clarified the statements as outlined above.

Minor considerations:

There are a few typos.

Results, p.8, 2nd-last line: "fluorescence"

Materials and Methods, p. 26, paragraph 2: "acetone"

Reply: We corrected these and a few other typos in the revised manuscript.

Reviewer #2 (Remarks to the Author):

The study “Deciphering the mechanism of glutaredoxin-catalyzed roGFP2 redox sensing reveals a ternary complex with glutathione for protein disulfide reduction” by Geissel et al. provides a detailed insight into the molecular mechanism and kinetics of the Grx-catalyzed oxidation and reduction of roGFP2. Although roGFP fusion constructs have a broad application in detecting intracellular redox potentials, the underlying enzyme mechanism that facilitate its reduction and oxidation was still not fully unraveled. Using glutaredoxin (Grx) from *Plasmodium falciparum* as model enzyme, the work provides a very detailed mechanistic insight into the oxidation and reduction of roGFP. Vice versa, oxidized roGFP(S2) served as an excellent tool to analyze the mechanism of the glutathione/glutaredoxin system in reducing non-glutathione disulfides. In the first part of the manuscript, the authors determined oxidation kinetics for reduced roGFP2WT (wild-type) in comparison to the single cysteine mutants, roGFP2C125S and roGFP2C208S and clearly showed that PfGrx efficiently catalyzes the oxidation of roGFP via a glutathione-dependent monothiol mechanism. The fusion construct between roGFP2WT and PfGrx variants further accelerates roGFP oxidation. Although this connection has been reported previously, the authors provide here the underlying kinetics of the reaction mechanism under physiological glutathione/glutathione disulfide concentration. The second part focuses on the PfGrx-catalyzed reduction of roGFP2. The authors determined the reduction kinetics for PfGrx-mediated deglutathionylation roGFP via a monothiol mechanism and further analyzed the reduction kinetics of roGFPWT(S2) as a non-glutathione disulfide substrate. Conducting detailed analyses of the reactions between reduced PfGrx variants and roGFPWT(S2), they revealed that the reduction follows a monothiol mechanism under formation of a ternary complex with PfGrx and GSH.

The article is well-written and the observed kinetic data is well-explained in the text. I particularly appreciated the schematic representations included into individual figures to illustrate the mechanistic models, which helped to understand their conclusion. Overall, the work represents a carefully conducted and very detailed kinetic analyses. The molecular mechanisms allow to better understand and evaluate intracellular studies based on roGFP as a redox sensor and provided novel mechanistic insight into the role of Grx in catalyzing the reduction of non-glutathione disulfides in the cell.

Reply: We thank the reviewer for his/her careful review and the kind summary of our study.

I have few minor comments that should be addressed by the authors:

In the abstract, the authors highlight that they initially analyzed PfGrx-catalyzed oxidation of roGFP2. Consistently, the “Results” part starts with the respective experiments. However, in the introduction, the authors did not specifically address the role of Grx in catalyzing the oxidation of protein substrates, such as roGFP2, but mainly focus on Grx-catalyzed reduction.

Reply: We added the following sentences to highlight the Grx-catalyzed oxidation of substrates in the introduction as well: "Since all reactions in Fig. 1 are reversible, class I glutaredoxins can also catalyze the GSSR- or GSSG-dependent glutathionylation of thiols or the formation of (protein) disulfides.^{3,5,35,61} Such oxidations are in direct competition with glutathione reductase and ABC

transporters which usually maintain very low GSSG concentrations under physiological conditions.^{5,7,62-64} We also modified the following sentence accordingly: "One exception is the analysis of fluorescent protein substrates such as rxYFP or roGFP2, which allow a direct continuous detection of the reversible glutaredoxin-dependent protein disulfide reduction and formation *in vitro*.^{14,15,31-35}" The reversibility of Grx-catalyzed redox reactions is now mentioned six times in the introduction.

Regarding Figure 2a, it was not clear to me, why roGFP2WT (SH)₂ was incubated with PfGrxC32S/C88S(SSG) to monitor oxidation, but both roGFP2 mutant variant, roGFP2C215S(SH) and roGFP2C208S(SH), were both treated with GSSG in the absence of Grx.

Reply: While the spectral changes for roGFP2^{WT} were already reported in the literature, we had no information regarding the position of the equilibrium for the reaction between monothiol His-tagged roGFP2(SH) and His-tagged PfGrx^{C32S/C88S}(SSG) and therefore chose to use an excess of GSSG (10 mM) for the glutathionylation of monothiol roGFP2 followed by the removal of GSSG and GSH by Ni-NTA affinity chromatography. Omitting PfGrx^{C32S/C88S} in these experiments also minimized the number of confounding parameters that might have affected the rather small spectral changes for roGFP2^{C151S} and roGFP2^{C208S}. Subsequent experiments with PfGrx^{C32S/C88S}(SSG) showed that the spectral changes for both monothiol roGFP2 variants were also reproducible using PfGrx^{C32S/C88S}(SSG) for glutathionylation (e.g., Fig. 2b).

p 8, l. 11: The authors further claim that the slight changes detected for glutathionylated roGFP2C215S and roGFP2C208S are significant, but they do not provide any statistical test to confirm their statement.

Reply: We did not perform any statistical analysis and changed the sentence accordingly to emphasize that the spectral changes for roGFP2^{C151S} and roGFP2^{C208S} were highly reproducible: "To address the reactivities of the redox-sensitive cysteinyl residues, we also analyzed the recombinant monothiol roGFP2 variants roGFP2^{C151S} and roGFP2^{C208S} and detected small but reproducible glutathionylation-dependent changes of fluorescence at both excitation wavelengths for roGFP2^{C151S}(SSG) and roGFP2^{C208S}(SSG) (Fig. 2a)."

In this regard, I do see the small changes in the spectrum of roGFP2C208 upon GSSG treatment as reflected by an increased fluorescence at an excitation of 400 nm and a decrease at 484 nm (Figure 2a, right). However, I do not see the same behavior for roGFP2C151S. If there is any significant change, the fluorescence decreased at both excitation wavelength upon GSSG treatment (Figure 2a, middle). Also, the stopped-flow measurements showed the decrease at both wavelength upon glutathionylation of roGFP2C151S (Figure 2b, middle) in contrast to roGFP2WT and roGFP2C208S. In the text, the authors did not address this observation, but instead write that "the spectral changes at both excitation wavelengths followed identical kinetics [...] for the formation of roGFP2WT(S₂), roGFP2C151S(SSG), or roGFP2C208S(SSG) using PfGrxC32S(C88S)(SSG) as an oxidant" (p8, l. 17-19). From the data shown in Figure 2a and b for roGFP2C151S, I do not understand how this conclusion has been drawn and would require some more explanation.

Reply: Yes, the small but reproducible change of fluorescence is different (i.e., negative) for roGFP2^{C151S} at an excitation of 400 nm. However, even though the exact underlying structural changes for the unusual spectrum of roGFP2^{C151S}(SSG) remain to be characterized, the rate constant k_{obs} does not depend on whether the change of fluorescence is positive or negative. The rate constants from Fig. 2b were identical regardless whether the kinetic fit was performed for the trace at 400 nm or at 484 nm (in other words, "the spectral changes at both excitation wavelengths followed identical kinetics"). Regarding the different spectral changes, we now added the following sentence for clarification: "While glutathionylation of roGFP2^{C208S} also caused an increased fluorescence at an excitation of 400 nm and a decreased fluorescence at an excitation of 484 nm, glutathionylation of roGFP2^{C151S} resulted in a decreased fluorescence at both excitation wavelengths."

REVIEWERS' COMMENTS

Reviewer #2 (Remarks to the Author):

The authors carefully revised the manuscript and addressed all points that have been initially raised by the reviewers. My questions have been considerably answered and the changes of the text provide now more clarity and further improve the understanding of this comprehensive mechanistic study, which I clearly recommend for publication in Nature Communications.